# Engineering cardiolipin binding to an artificial membrane protein reveals determinants for lipid-mediated stabilization

Mia L Abramsson[1], Robin A Corey[2]*, Jan L Skerle[3,4], Louise J Persson[5], Olivia Anden[6], Abraham O Oluwole[7,8], Rebecca J Howard[6], Erik Lindahl[6,9], Carol V Robinson[7,8], Kvido Strisovsky[4], Erik G Marklund[5], David Drew[3], Phillip J Stansfeld[10]*, Michael Landreh[11]*

[1]Department of Microbiology, Tumor and Cell Biology, Karolinska Institutet, Solna, Sweden; [2]School of Physiology, Pharmacology & Neuroscience, University of Bristol, Bristol, United Kingdom; [3]Department of Biochemistry and Biophysics, Stockholm University, Stockholm, Sweden; [4]Institute of Organic Chemistry and Biochemistry, Academy of Science of the Czech Republic, Prague, Czech Republic; [5]Department of Chemistry – BMC, Uppsala University, Uppsala, Sweden; [6]Department of Biochemistry and Biophysics, Science for Life Laboratory, Stockholm University, Solna, Sweden; [7]Department of Chemistry, University of Oxford, Oxford, United Kingdom; [8]Kavli Institute for Nanoscience Discovery, University of Oxford, Oxford, United Kingdom; [9]Department of Applied Physics, Science for Life Laboratory, KTH Royal Institute of Technology, Solna, Sweden; [10]School of Life Sciences & Chemistry, University of Warwick, Coventry, United Kingdom; [11]Department for Cell and Molecular Biology, Uppsala University, Uppsala, Sweden

*For correspondence:
robin.corey@bristol.ac.uk (RAC);
phillip.stansfeld@warwick.ac.uk
(PJS);
michael.landreh@ki.se (ML)

Competing interest: The authors declare that no competing interests exist.

## eLife Assessment

Cardiolipin is known to play an **important** role in modulating the assembly and function of membrane proteins in bacterial and mitochondrial membranes. Here, authors **convincingly** define the molecular determinants of cardiolipin binding on de novo-designed and native membrane proteins combining the coarse-grained molecular dynamics simulation with the state-of-the-art experimental approaches such as native mass spectrometry and cryogenic electron microscopy. The major findings in this study, which are the identification of degenerate cardiolipin binding motifs, the characterization of their dynamic features, and the role in membrane protein stability and activity, will provide much needed insight into the still poorly understood nature of protein-cardiolipin interactions.

**Abstract** Integral membrane proteins carry out essential functions in the cell, and their activities are often modulated by specific protein-lipid interactions in the membrane. Here, we elucidate the intricate role of cardiolipin (CDL), a regulatory lipid, as a stabilizer of membrane proteins and their complexes. Using the in silico-designed model protein TMHC4_R (ROCKET) as a scaffold, we employ a combination of molecular dynamics simulations and native mass spectrometry to explore the protein features that facilitate preferential lipid interactions and mediate stabilization. We find that the spatial arrangement of positively charged residues as well as local conformational flexibility

are factors that distinguish stabilizing from non-stabilizing CDL interactions. However, we also find that even in this controlled, artificial system, a clear-cut distinction between binding and stabilization is difficult to attain, revealing that overlapping lipid contacts can partially compensate for the effects of binding site mutations. Extending our insights to naturally occurring proteins, we identify a stabilizing CDL site within the *E. coli* rhomboid intramembrane protease GlpG and uncover its regulatory influence on enzyme substrate preference. In this work, we establish a framework for engineering functional lipid interactions, paving the way for the design of proteins with membrane-specific properties or functions.

## Introduction

Biological membranes, which are vital for cellular life, provide a specific and highly adaptable lipid environment for membrane proteins that govern numerous cellular functions (*Levental and Lyman, 2023*). The exact roles that the different membrane lipids play in the regulation of membrane proteins often go unacknowledged, as their highly dynamic interactions challenge conventional analytical methods. Despite these obstacles, evidence has consistently highlighted the crucial role of lipids (*Corradi et al., 2019*), for example as allosteric regulators (*Cournia and Chatzigoulas, 2020*), facilitating protein oligomerization (*Renard and Byrne, 2021*), or locally affecting the properties of the membrane (*Bozelli et al., 2021*). The simplest form of lipid-mediated regulation is the stabilization of specific protein conformations (*Hunte, 2005*), resulting in the observation of individual lipid molecules in high-resolution structures (*Palsdottir and Hunte, 2004*). These 'structural' lipids often display increased residence times at their binding sites which distinguish them from non-regulatory, 'annular' lipids (*Landreh et al., 2016*).

Cardiolipin (CDL) is a prime example of a lipid with regulatory activity for both bacterial and mitochondrial membrane proteins (*Musatov and Sedlák, 2017*). Due to its unique structure, comprised of two phosphate groups which both potentially carry a negative charge, and four acyl chains, CDL mediates the assembly of membrane protein oligomers, for example in the respiratory chain supercomplexes (*Pfeiffer et al., 2003*). The double phosphate groups can create strongly attractive electrostatic interactions with basic side chains, which makes CDL an idea model lipid to understand interactions, but it also exhibits more specific patterns. Of note, both the head groups and all four acyl chains are thought to be important components of supercomplex stabilization (*Corey et al., 2022*). Similarly, CDL plays an essential role in the dimerization of the $Na^+/H^+$ antiporter NhaA, which increases the exchanger activity to protect the bacteria from osmotic stress (*Landreh et al., 2017*; *Rimon et al., 2019*). In addition, CDL can affect the activity of other membrane proteins such as ADP/ATP carrier Aac2 and magnesium transporter MgtA by acting as an allosteric regulator (*Senoo et al., 2024*; *Weikum et al., 2024*). Therefore, sites displaying preferential CDL binding may indicate lipid-activated regulatory mechanisms. To address this possibility, we have previously used coarse-grained molecular dynamics (CG-MD) simulations to map CDL binding sites on *E. coli* inner membrane proteins with published structures, identifying specific amino acids and binding site geometries that mediate preferential interactions with CDL (*Corey et al., 2021*). Although such CDL 'fingerprints' are found in a wide range of proteins with different activities, they stop short of clarifying the functional role of lipids at these sites, with predictions of their functionality remaining largely speculative. Addressing this knowledge gap requires monitoring both the molecular interactions as well as the structure or stability of membrane protein complexes. For instance, thermal-shift assays provide data on lipid binding and associated changes in protein stability, which may indicate a functionally or structurally important lipid interaction (*Nji et al., 2018*). Moreover, native mass spectrometry (nMS) has gained traction for membrane protein analysis, revealing the influence of lipids on oligomerization (*Gupta et al., 2017*), binding affinities (*Schrecke et al., 2021*), and conformational stability (*Laganowsky et al., 2014*). Monitoring mass shifts captures individual lipid interactions across multiple protein populations, while gas-phase dissociation provides insight into lipid stabilization. nMS thus captures key features of regulatory lipid interactions, and is especially powerful when coupled with MD which provides insight at the atomistic level (*Bolla et al., 2020*).

Being able to connect individual lipid binding events to the stability of a protein complex is a crucial step toward predicting functionally important CDL interactions. We reasoned that a combined MD and nMS strategy may reveal basic requirements for CDL-mediated stabilization. However, the

sequence and structures of membrane proteins are evolutionarily entrenched with the lipid composition of their surrounding membrane. To reduce the system to first principles, we turned to Trans-Membrane Helical Core Tetramer_Rocket-shaped (TMHC4_R, hereafter referred to as ROCKET), an artificial membrane protein tetramer whose sequence was derived from Rosetta Monte Carlo calculations (*Lu et al., 2018*). ROCKET includes a generic lipid-water interface composed of a ring of aromatic residues and a ring of positively charged residues on the cytoplasmic side. Into the ROCKET scaffold, we designed several CDL binding sites based on our observations from *E. coli* proteins and tested their effect on tetramer stability using nMS. We find that local dynamics and the spatial distribution of charged residues distinguish stabilizing from non-stabilizing sites. However, we also observe that predicting the impact of individual mutations on lipid binding and stabilization from the structure can be challenging, even in our highly artificial system. These difficulties arise from the fact that lipid interactions are heterogeneous, and the loss of one type of contact may be compensated by another. Screening our database of *E. coli* CDL binding sites (https://osf.io/gftqa/) for binding sites that resemble stabilizing sites in ROCKET, we uncover a highly stabilizing CDL interaction in the membrane protease GlpG, which regulates the substrate preference of the enzyme. In summary, our study demonstrates the potential as well as the challenges in designing functional CDL sites on artificial proteins that can recognize membrane compositions.

## Results

### Design of a CDL binding site in ROCKET

As first step, we characterized the inherent lipid binding properties of ROCKET (*Figure 1a*) through CG-MD simulations of the protein in a mimetic *E. coli* membrane. The membrane composition was modeled with a distribution of POPE, POPG, and CDL in a ratio of 67:23:10, and the system was simulated for 5x10 µs while monitoring the lipid interactions. We observed abundant lipid interactions, with CDL displaying markedly more localized binding than POPE or POPG (*Figure 1—figure supplement 1*). The N-terminal region on the first transmembrane helix bound CDL with average occupancy of 71% and average residence time of 35 ns (at R9). These values are extracted from the full 50 µs of simulation data. The site, which we termed Site 1, consists of three basic residues (R9, K10, and R13) and an aromatic residue (W12), which corresponds to a consensus CDL binding motif (*Figure 1b*; *Corey et al., 2021*). We also observed a second site, involving W12 in a slightly rotated conformation, and R66 on helix 2 of the neighboring subunit. This site, termed Site 2, exhibited significantly lower occupancy of 56% and an average residence time of 35 ns (at R66). W12 can engage in CDL binding at either site, including simultaneously both sites. Both sites represent distinct lipid binding modes: Site 1 is a high-occupancy site away from the protein core with extensive head-group interactions, and Site 2 is a lower-occupancy site with extensive acyl chain contacts close to the protein core. Note that, while the occupancies are high, the residence times are relatively low, as CDL is readily exchanged between the two sites. We decided to use these two sites, which arose from purely statistical distribution of charged and aromatic residues, as basis for engineering a stabilizing CDL site.

To separate the two CDL binding modes, we generated ROCKET mutants in silico and performed CG-MD with two subunits each of ROCKET and ROCKET mutants. We found that substituting the charged residues of Site 1 with alanine (R9A/K10A/R13A, *Figure 1c*) redirected preferential CDL binding to Site 2 (*Figure 1d and e*). In this mutant, which we termed ROCKET[AAXWA], the Site 2 had an occupancy of 53% and an increased average residence time of 47 ns (R66), whereas Site 1 had a reduced occupancy of 45% and a residence time of 45 ns (R9A). This occupancy difference was quantified by CG simulations and showed significant reduction of total CDL binding between ROCKET and ROCKET[AAXWA] (*Figure 1d*, *Figure 1—figure supplement 1*).

Next, we evaluated the potential for lipid-mediated stabilization at both sites using gas-phase atomistic MD simulations, which allows for a direct comparison with nMS. We applied a pulling force between two adjacent subunits of ROCKET and ROCKET[AAXWA] tetramers with and without bound CDL and determined the force required to separate the protein chains (*Figure 1f*). Analysis of the secondary structure content shows that the AAXWA mutation stabilizes the conformation of helix 1. However, we found that separating the adjacent helices from the neighboring subunits of ROCKET[AAXWA] by 1.1 nm, the point at which non-covalent interactions between the transmembrane helices are disrupted, required more force when CDL was present (*Figure 1g*). ROCKET, on the other hand,

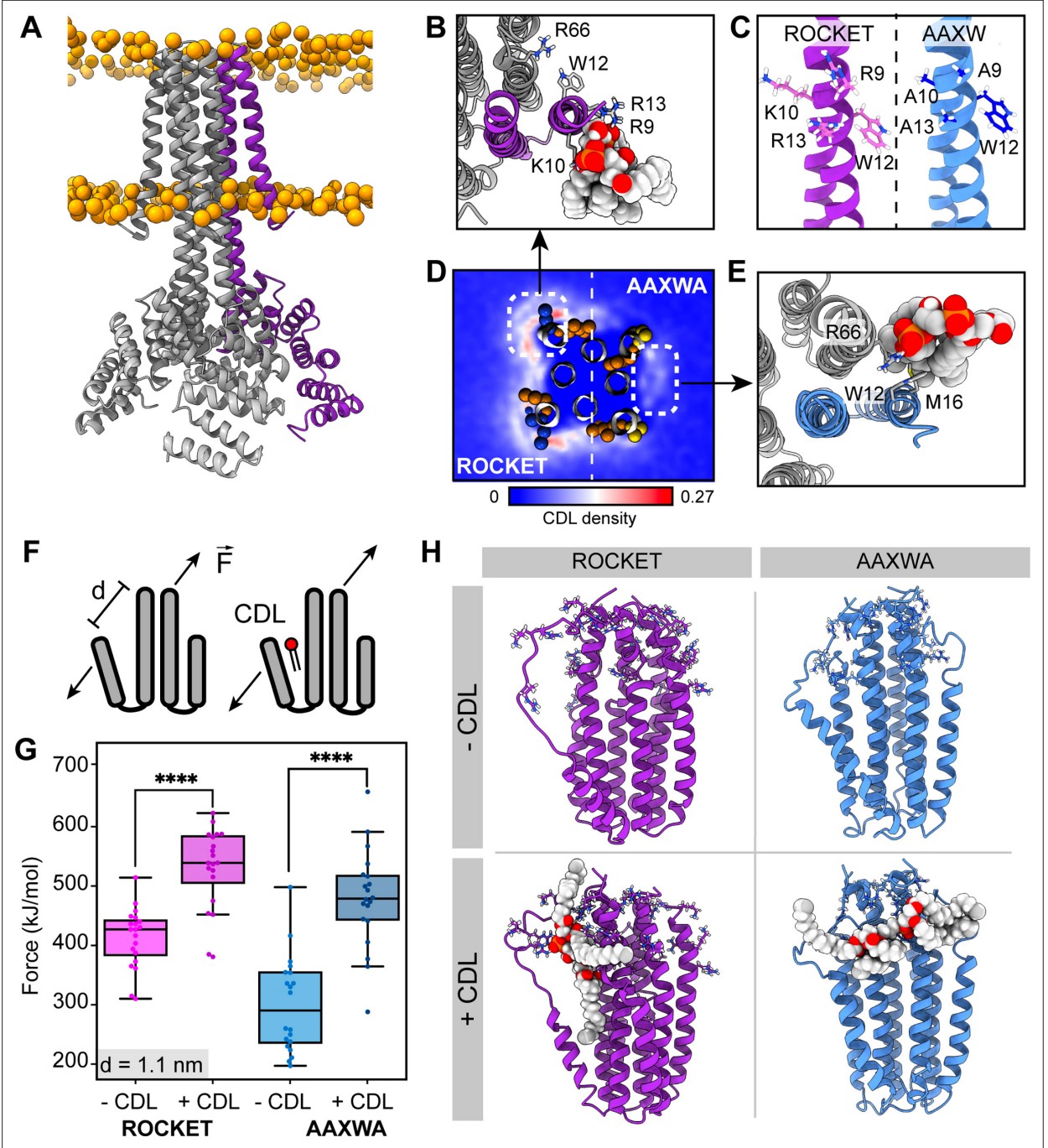

**Figure 1.** ROCKET contains two CDL binding sites with different structural implications. (**A**) Structure of ROCKET (PDB ID 6B85) in the membrane, with one protein subunit highlighted in purple and phosphate headgroups for the membrane shown as orange spheres. The structure was obtained from MemProtMD. (**B**) Top view of CDL binding to Site 1 taken from a 10 µs snapshot of a CG-MD simulation. The poses was converted to atomistic using CG2AT2 (*Vickery and Stansfeld, 2021*) to the CHARMM36m force field (*Huang and MacKerell, 2013*). CDL is shown as spacefill, and residues R9, K10, W12, R13, and R66 as sticks. The CDL-binding subunit is highlighted in purple. (**C**) Design of the ROCKET[AAXWA] variant. Site 1 on helix 1 of ROCKET is shown on the left (purple), and ROCKET[AAXWA] with the mutations R9A/K10A/R13A on the right blue. (**D**) CG-MD-derived CDL densities around a heterotetramer composed of two ROCKET subunits (left) and two ROCKET[AAXWA] subunits (right). Units are number density. Site 1 on ROCKET and Site 2 on ROCKET[AAXWA] are highlighted by dashed boxes. R9, K10, W12, and R13 are shown as spheres (basic in blue, aromatic in orange). Densities are computed over 5x10 µs simulations. (**E**) Top view of CDL binding to Site 2 in ROCKET[AAXWA] following CG-MD and converted to atomistic as per panel B. Interacting residues W12, M16, and R66 on the neighboring subunit (grey) are shown as sticks. (**F**) Setup of gas-phase MD simulations for unfolding of ROCKET and ROCKET[AAXWA] with and without lipids. The placement of the lipid in the schematic is arbitrary. (**G**) Plots of the integral of the force required to separate helix 1 and 2 (d=1.1 nm), for ROCKET (purple) ($p=3.85*10^{-7}$) and ROCKET[AAXWA] (blue) ($p=2.93*10^{-8}$) with and without bound CDL show a

*Figure 1 continued on next page*

*Figure 1 continued*

more pronounced increase in stability for CDL-bound ROCKET^AAXWA compared to ROCKET (two-tailed t-test with n=20).(**H**) Snapshots from gas-phase MD simulations show broad interactions of CDL across the subunits of lipid-bound ROCKET^AAXWA (blue) and more localized interactions with fewer intermolecular contacts for ROCKET (purple). Amino acid position 9, 10, 12, 13, and 66 are shown as sticks in each subunit.

The online version of this article includes the following figure supplement(s) for figure 1:

**Figure supplement 1.** CG-MD simulations of lipid interactions with ROCKET and ROCKET^AAXWA.

**Figure supplement 2.** Molecular dynamics simulations of ROCKET in the gas phase.

displayed a lower significant difference in force with or without CDL. Snapshots from the simulations reveal that the lipid forms multiple contacts with both subunits adjacent to Site 2 in ROCKET^AAXWA, which likely gives rise to the stabilizing effect (*Figure 1h*). We conclude that channeling the CDL molecules to inter-helix sites may be a prerequisite for lipid-mediated stabilization.

## Inter-helix CDL binding stabilizes ROCKET^AAXWA in the gas-phase

Having derived two ROCKET variants with distinct CDL binding modes from MD simulations, we turned to cryogenic electron microscopy (cryo-EM) and nMS to investigate their lipid interactions experimentally. We first analyzed ROCKET and ROCKET^AAXWA in the presence of CDL by cryo-EM. The resulting density maps for each protein with no other particle class detected, refined to a resolution of 3.8 and 3.9 Å respectively, show essentially identical architectures that agree with the previously solved crystal structure of ROCKET (*Lu et al., 2018*), confirming that the mutations do not disrupt the native structure (*Figure 2—figure supplement 1*, *Supplementary file 1, table S1*). Although a definitive atomic-level molecular model was not possible at this resolution, we also observed in both maps a diffuse non-protein density which partially overlaps the head-group of CDL predicted in Site 2 (*Figure 2—figure supplement 1*). Interestingly, we see no extra density in Site 1, however, this site is more exposed, making it more likely that excess detergent can outcompete the binding of CDL in this site. Furthermore, the orientation of CDL is more flexible in Site 1 than Site 2 (*Figure 1D*), which also reduces the likelihood of obtaining a sufficiently defined density. To determine lipid binding preferences, we therefore reconstituted the proteins into liposomes composed of polar *E. coli* polar lipid extracts (*Figure 2a*). By releasing the proteins from the liposomes inside the mass spectrometer and monitoring the intensity peaks corresponding to apo- and lipid-bound protein, we can compare lipid preferences of both variants (*Figure 2a*). We find that tetrameric ROCKET retains up to three CDL molecules, which can be identified by their characteristic 1.4 kDa mass shift, as well as a significant number of phospholipids between 700 and 800 Da (*Figure 2b*). The data thus show a preference for CDL, which constitutes only 10% of the liposome. Interestingly, nMS of ROCKET^AAXWA revealed a similarly specific retention of up to three CDL molecules for 17+charge state, although the intensity of the lipid adducts was reduced by approximately 50% (*Figure 2c*). The mass spectra show that the preference for CDL is preserved in the ROCKET^AAXWA variant, while the occupancy is reduced, indicating either lower affinity in solution or lower stability of the protein-lipid complex in the gas-phase.

To validate these findings, we analyzed lipid binding to detergent-solubilized ROCKET. First, we optimized detergent conditions for maintaining the intact ROCKET tetramer (*Figure 2—figure supplement 2*). Importantly, we observe no co-purified lipids, indicating that detergent can remove CDL during extraction, as expected for lipids with short residence times (*Bolla et al., 2020*). We then performed a competition assay where we mixed both variants in C8E4 detergent containing a limiting amount of CDL (50 μM) and monitored binding with nMS. As expected, both variants bound three distinguishable CDL molecules per tetramer; however, ROCKET displayed significantly more intense lipid adducts than ROCKET^AAXWA (*Figure 2—figure supplement 3*). In CG-MD simulations, the overall CDL occupancy is lower in ROCKET^AAXWA than in ROCKET, meaning fewer lipids will be bound simultaneously. The nMS data show CDL retention by both variants, but the ROCKET^AAXWA protein has lower-intensity CDL adduct peaks (*Figure 2B and C*). This finding suggests that both variants bind CDL, but in the ROCKET^AAXWA variant, the sites have lower occupancy. The nMS data are therefore consistent with CDL binding preferentially to Site 1 in ROCKET and preferentially to Site 2 in the ROCKET^AAXWA variant.

Next, we explored how CDL binding to either site affects the stability of ROCKET, using the oligomeric state in nMS as a measure. To avoid interference from different lipids in the reconstituted

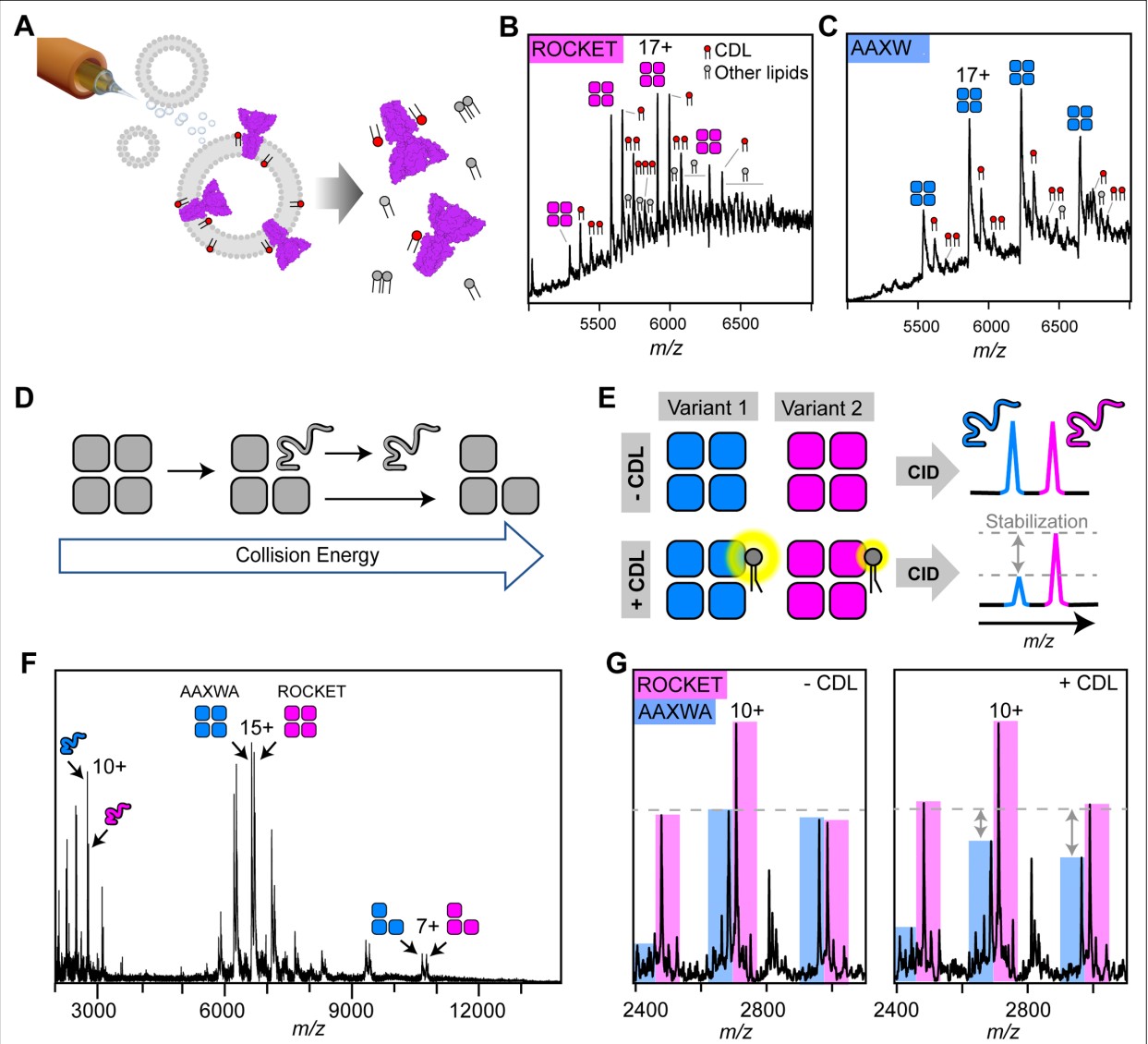

**Figure 2.** nMS analysis of lipid binding and lipid mediated stabilization of ROCKET and ROCKET[AAXWA]. (**A**) Schematic depiction of electrospray ionization (ESI) process for proteo-liposomes, leading to the ejection of protein-lipid complexes into the gas-phase. (**B**) A representative mass spectrum of ROCKET released from proteoliposomes shows tetramers with 1–3 bound CDL molecules, as judged by the characteristic mass shift of 1.4 kDa, as well as additional lipids with molecular weights between 700 and 800 Da. (**C**) Release of ROCKET[AAXWA] from proteoliposomes shows retention of CDL molecules. For the 18+ion of the tetramer, a maximum of three CDL adducts can be assigned unambiguously. The reduced lipid adduct intensity compared to ROCKET indicates reduced lipid binding and/or complex stability. (**D**) Schematic illustrating the process of gas-phase subunit unfolding and ejection from ROCKET tetramers at increasing collision energies. (**E**) A nMS assay to assess CDL-mediated stabilization of ROCKET and ROCKET[AAXWA]. Simultaneous dissociation of ROCKET and ROCKET[AAXWA] leads to the ejection of unfolded monomers, which can be quantified by nMS (top row). Addition of CDL to the same mixture results in lipid binding to tetramers. If CDL binding stabilizes one tetrameric variant more than the other, the amount of ejected monomers will be reduced accordingly (bottom row). (**F**) Representative mass spectrum of ROCKET and ROCKET[AAXWA] at a collision voltage of 220 V. Intact tetramers are seen in the middle, ejected monomers and stripped trimers are seen in the low and high m/z regions, respectively. (**G**) Zoom of the low m/z region of a mixture of 25 μM each of ROCKET and ROCKET[AAXWA] with a collision voltage of 200 V before (left) and after (right) the addition of 50 μM CDL. The three main charge states for both variants can be distinguished based on their mass difference. Addition of CDL reduces the intensity of the ROCKET[AAXWA] monomer peaks compared to ROCKET (dashed line).

The online version of this article includes the following figure supplement(s) for figure 2:

**Figure supplement 1.** Cryo-EM densities for ROCKET and ROCKET[AAXWA] in the presence of CDL.

**Figure supplement 2.** Detergent screening and CDL binding to ROCKET and ROCKET[AAXWA].

**Figure supplement 3.** nMS spectra showing lipid-mediated stabilization of ROCKET[AAXWA] and ROCKET[R66A] (yellow).

liposome system, we switched to detergent micelles as vehicles for nMS and employed gas-phase dissociation of the intact protein complexes to remove bound detergent. Briefly, collisions with gas molecules in the ion trap of the mass spectrometer cause thermal unfolding of a single subunit in the complex, which is then ejected as a highly charged, unfolded monomer (*Figure 2d*; *Hyung et al., 2009*). By comparing the peak intensities of the monomers that are ejected simultaneously from two protein oligomers, we can obtain information about their relative stabilities. Therefore, by adding CDL to an equimolar mixture of ROCKET and ROCKET$^{AAXWA}$, dissociating the resulting complexes, and monitoring changes in monomer signal intensities, we can determine whether lipid binding to Site 1 or Site 2 affects tetramer stability (*Figure 2e*). Importantly, by comparing changes in peak intensities with and without CDL while keeping all other conditions constant, we can avoid interference from changes in gas-phase fragmentation or ionization efficiency. nMS of ROCKET and ROCKET$^{AAXWA}$ shows the release of highly charged monomers which can be distinguished based on their masses (*Figure 2f*). We then added CDL to the protein solution and repeated the measurement using identical conditions. We observed a reduction in the peak intensities of ROCKET$^{AAXWA}$ monomers compared to ROCKET (*Figure 2g*). We do not observe a change in the charge state distributions for tetramers or monomers, or notable fragmentation. Therefore, the change in monomer ratio suggests that CDL stabilizes the ROCKET$^{AAXWA}$ tetramer to a greater extent than the ROCKET tetramer. These findings are surprising, since the AAXWA variant displays significantly lower lipid binding (*Figure 2B and C*). However, considering the predictions from CG- and gas-phase MD, the increase in stability can be attributed to the preferential binding of CDL to the inter-helix Site 2 in the AAWXA variant. CDL binding to the distal Site 1, as preferred in ROCKET, involves fewer intermolecular contacts, and is therefore unlikely to exhibit a similarly stabilizing effect.

## Multiple structural features impact CDL-mediated stabilization

The finding that inter-helix CDL binding stabilizes a tetrameric membrane protein in the gas-phase recapitulates a key feature of both prokaryotic and eukaryotic membrane proteins (*Gupta et al., 2017*; *Pyle et al., 2018*). Unlike naturally evolved proteins, however, the extraordinary stability and mutation tolerance of the ROCKET scaffold enables us to dissect further the requirements for CDL-mediated stabilization. We therefore applied the above MS strategy to quantitatively assess lipid-mediated stabilization in our model system between two protein variants using ROCKET$^{AAXWA}$ as an internal reference. We can determine relative stability changes upon CDL addition for different ROCKET variants by plotting the ratios of the total intensity of the peaks for monomeric ROCKET mutants (ROCKET$^{MUT}$) to the total intensity of all protein monomer peaks in the spectrum (ROCKET$^{MUT}$ +ROCKET$^{AAXWA}$) with and without CDL. If ROCKET$^{AAXWA}$ is stabilized more than the variant of interest, the ratio increases with CDL addition (*Figure 3a*). As expected, the AAXWA mutation significantly increased the stabilizing effect of CDL, as determined from four independent repeats (*Figures 2g, 3b and c*). With this assay, we then explored whether introducing different structural features into Site 1 could turn it into a stabilizing CDL binding comparable to Site 2. As a first hypothesis, we reasoned that a destabilization of the core of ROCKET might increase the effect of CDL. We introduced a destabilizing mutation (A61P) in helix 2, right below the headgroup region, theorizing that the proline-induced kink would destabilize the ROCKET tetramer. AlphaFold2 predictions (*Jumper et al., 2021*; *Akdel et al., 2022*) indicated that the ROCKET$^{A61P}$ mutation does not affect the tetrameric state (*McBride et al., 2023*), which was confirmed by nMS. To our surprise, ROCKET$^{A61P}$ exhibited significantly less CDL stabilization than ROCKET$^{AAXWA}$, and was comparable to ROCKET (*Figure 3d*, *Figure 2—figure supplement 3*). This observation indicates that the introduction of a proline in the protein core does not sufficiently destabilize the protein to be counteracted by lipid binding.

As second hypothesis, we reasoned that additional lipid binding sites may increase the effect of CDL binding (*Figure 3—figure supplement 1*). We therefore mutated residue D7 to alanine and S8 to arginine. The D7A/S8R variant retains the high-affinity Site 1 on helix 1, but includes an additional site composed of R8 on helix 1 and R66 and E68 on helix 2 which does not overlap with Site 1 (*Figure 3e*). Quantification of the monomer release with and without CDL suggests a shift towards increased stability with CDL, albeit not as pronounced as for ROCKET$^{AAXWA}$ (*Figure 3d*, *Figure 2—figure supplement 3*). This data leads us to speculate that Site 1 may still bind the bulk of the available CDL molecules, or that the salt bridge S8-E68 may contribute increased stability in a CDL-independent manner.

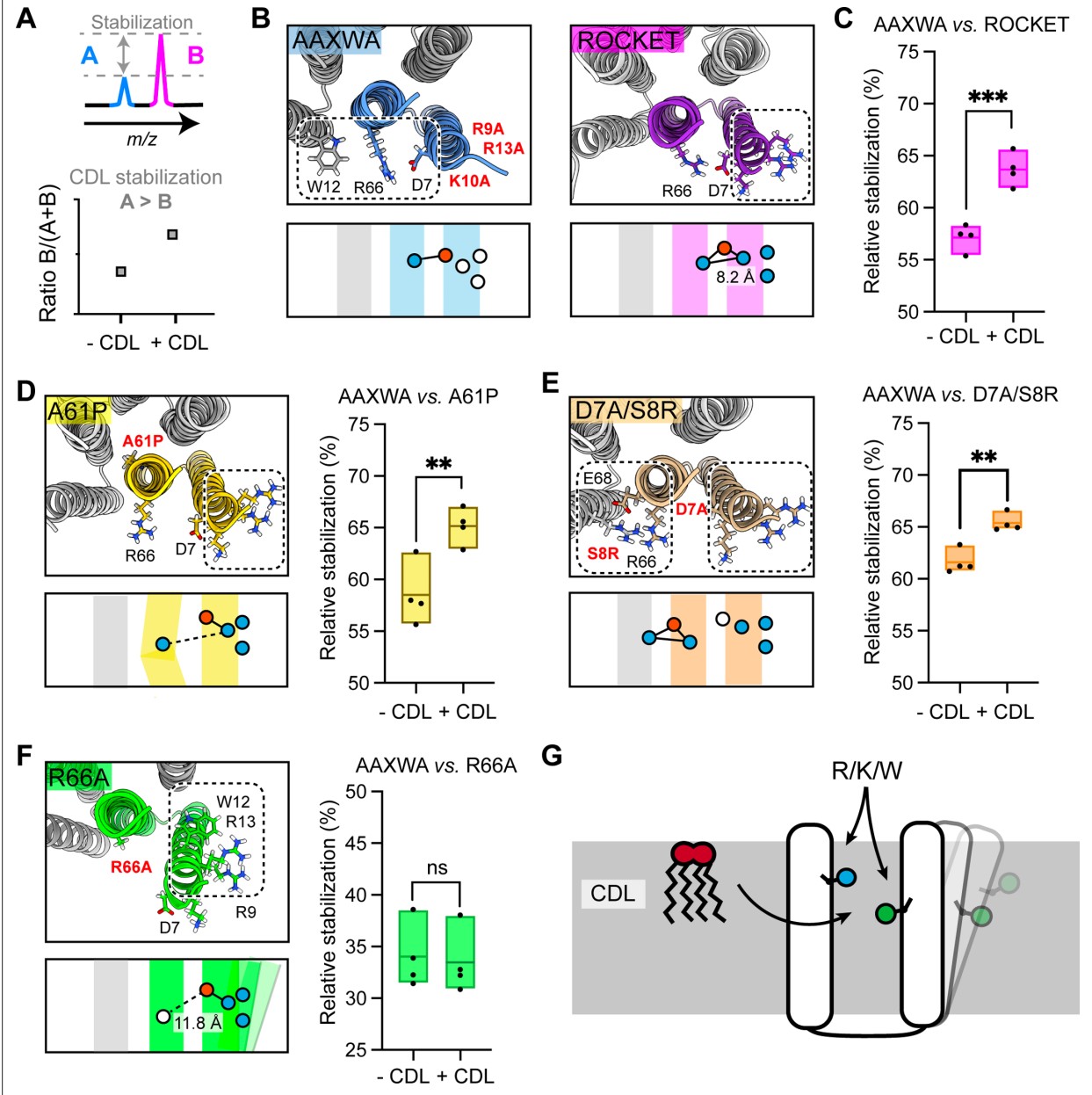

**Figure 3.** Assessment of lipid-mediated stabilization effects on ROCKET variant. (**A**) Principle for pairwise analysis of protein stabilization by CDL. Peaks representing monomers released of protein A and B display intensity changes upon lipid addition. Plotting the peak intensities as a ratio of B to total protein (A+B) shows an increase upon CDL addition, if A is stabilized more than B. (**B**) Residues involved in CDL headgroup binding to ROCKET and ROCKET[AAXWA] were derived from CG-MD simulations (***Figure 1***) and are shown based on AlphaFold2 models as top view, with the area occupied by CDL as a dashed rectangle. A single subunit is colored (purple for ROCKET and blue for ROCKET[AAXWA]). Below the structure, the orientation of the CDL site and the helices are shown as schematics. Positively and negatively charged residues are blue dots, respectively. Residues mutated to alanine are shown as white dots. (**C**) Plotting the peak intensity ratios of ROCKET to (ROCKET[AAXWA] +ROCKET) in the presence and absence of CDL shows a decrease in ROCKET[AAXWA] monomers when CDL is added (p=0.0007, two-tailed t-tests with n=4). (**D**) The CDL binding site and location of the A61P mutation (yellow) mapped on the AlphaFold2 model of ROCKET[A61P] and shown as a schematic as a side view below. Intensity ratios show significantly more pronounced stabilization of ROCKET[AAXWA] than ROCKET[A61P] (p=0.0084, two-tailed t-tests with n=4). (**E**) Introduction of a second CDL binding site in the D7A/S8R variant (orange) mapped on the AlphaFold2 model of ROCKET[D7A/S8R] and shown as a schematic as a side view below. The shift in intensity ratios show that ROCKET[AAXWA] is still stabilized to a greater extent (p=0.0019, two-tailed t-tests with n=4), but with a smaller margin than ROCKET or ROCKET[A61P]. (**F**) The R66A mutation, designed to disconnect helix 1 from the tetrameric protein core, results in an outward rotation of the CDL binding site, as shown in the AlphaFold2 model (green) and the side view schematic. Intensity ratios show no change upon CDL addition, suggesting that ROCKET[R66A] is stabilized to a similar extent as ROCKET[AAXWA] (p=0.8113, two-tailed t-tests with n=4). (**G**) Conceptual diagram depicting structural features that promote

*Figure 3 continued on next page*

*Figure 3 continued*

CDL-mediated stabilization. Distributing the residues that interact with the lipid headgroup, usually basic and aromatic residues, between two helices, as well as involvement of flexible protein segments, indicated by an outward movement of the right helix, also enhances stabilization by CDL.

The online version of this article includes the following figure supplement(s) for figure 3:

**Figure supplement 1.** Lipid binding to ROCKET^AAXWA vs ROCKET^D7A/S8R.

Having explored directed lipid binding (AAXWA), increased lipid binding (D7A/S8R) and core destabilization (A61P), we reasoned that the flexibility of the CDL binding site may affect stabilization. This feature is challenging to implement in the ROCKET scaffold, since it is designed around a tightly folded hydrogen bond network with a melting temperature of >90 °C (*Lu et al., 2018*). We therefore decided to untether helix 1 from the core of the protein by mutating R66, which forms a salt bridge with D7, to alanine. The AlphaFold2 model shows helix 1 being tilted away form the core, creating a large hydrophobic gap in the transmembrane region and turning Site 1 toward the neighboring subunit (*Figure 3f*). Interestingly, ROCKET^R66A showed lower signal intensities in native mass spectra than all other variants, which may indicate overall lower stability in detergent. Quantification of the monomer release with and without CDL revealed no significant difference compared to ROCKET^A-AXWA, which means CDL binding has a stabilizing effect on both proteins (*Figure 3f*). This finding is surprising, since loss of R66 should result in increased binding to Site 1 and thus not stabilize the protein. However, untethering helix 1 may create an opportunity for CDL coordinated by W12 and/or R9-R13 to insert its acyl chains into the resulting inter-helix gap. In this manner, CDL could exert a stabilizing effect in the absence of preferential headgroup interactions.

From the designed ROCKET variants, we can conclude that structure-based predictions of stabilizing CDL interactions is challenging, as they arise from a combination of headgroup- and acyl chain interactions, as well as from their impact on the local structural dynamics of the protein. However, from our observations, we can conclude that CDL binding involving different helices, as in ROCKET^AAXWA, and connecting flexible regions, as in ROCKET^R66A, gives rise to the most pronounced CDL stabilization of our system (*Figure 3g*). Core destabilization, as well as introduction of additional headgroup contacts, had less of an impact, although the specific properties of the engineered protein scaffold may mitigate potential effects to some extent.

## Identification of a stabilizing CDL binding site in the *E. coli* rhomboid intramembrane protease GlpG

As outlined above, the features that cause CDL-mediated stabilization of the ROCKET scaffold were designed based on observations from CG-MD investigation of CDL interactions with of *E. coli* membrane proteins. We therefore asked whether the same features could indicate stabilizing, and by extension, functionally relevant CDL interactions in naturally occurring proteins. To test this hypothesis, we evaluated our database of monomeric *E. coli* membrane proteins CDL sites (https://osf.io/gftqa/) for the sequence distribution of basic residues to find binding sites that span multiple helices. We reasoned that if two or more basic residues that interact with the same CDL molecule are located further apart in the sequence than approximately 30 positions, they have a high likelihood of being on separate helices, whereas sites spanning less than 10 residues are confined to a single helix or loop. Plotting the maximum sequence distance on a log scale reveals a bimodal distribution, with 75 CDL sites spanning less than 30 positions, and 180 sites spanning more than 30 positions (*Figure 4a*, *Figure 4—figure supplement 1*). Site 1 in ROCKET is in the first group, with four positions between R9 and R13. This approach does not consider interfacial CDL molecules in homo-oligomers, which may bind *via* single residues on different subunits. We therefore limited the dataset to monomeric proteins with CDL sites spanning >30 residues which we manually inspected to find CDL sites linking potentially flexible regions.

The rhomboid intramembrane protease GlpG from *E. coli* contains a CDL site between R92 on helix 1 and K167 on helix 4, as well as one aromatic residue on each (W98 and Y160). Importantly, R92 borders on a disordered linker region that connects helix 1 to an N-terminal cytoplasmic domain (*Figure 4b*). These features suggest that CDL binding may have structural implications. We therefore measured whether any lipids imparted thermal stabilization in a GFP-based thermal shift assay (*Nji et al., 2018*). Measuring the fraction of GFP-tagged soluble protein after heating to 63 °C in the

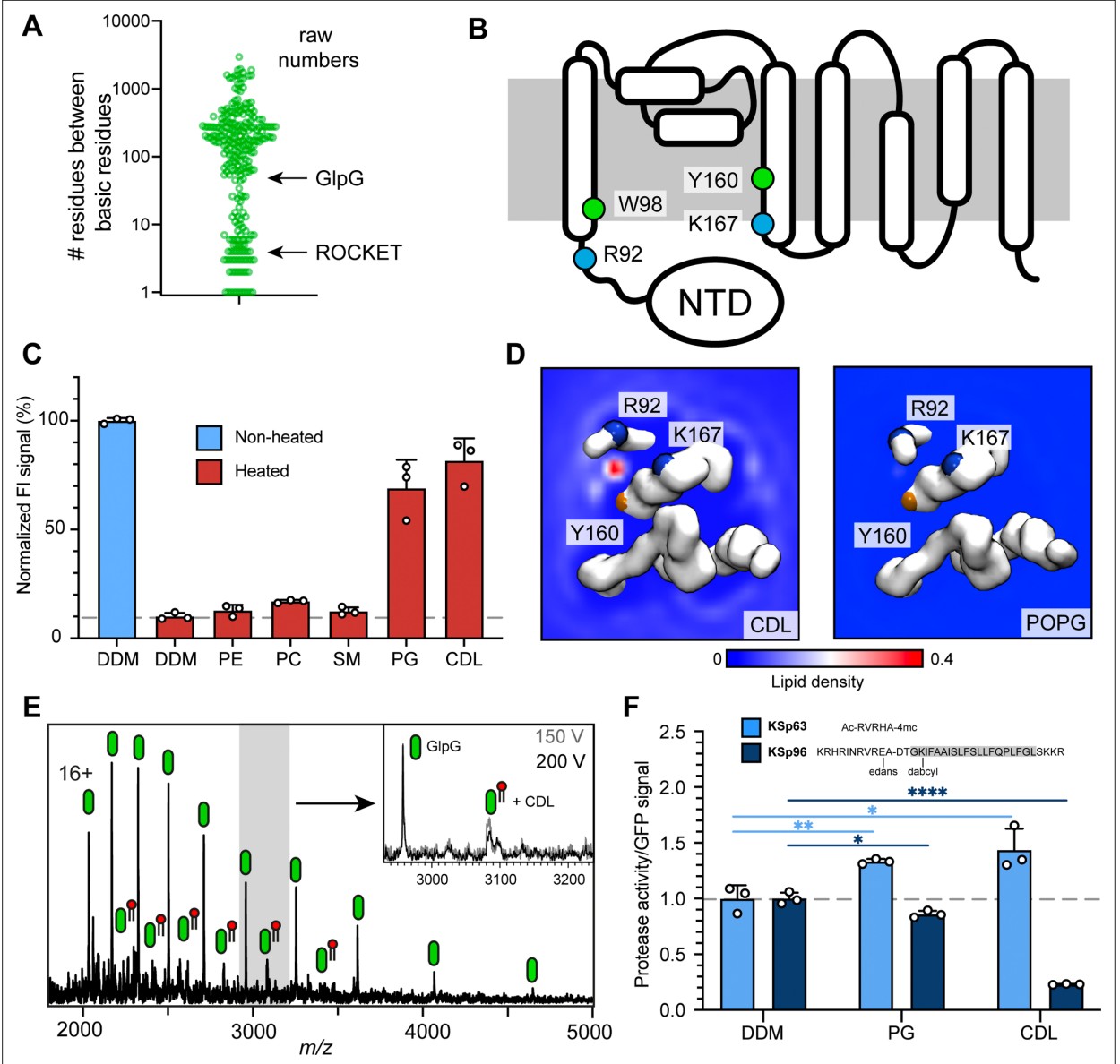

**Figure 4.** Identification of a functional CDL binding site in the GlpG membrane protease. (**A**) Distances between basic residues within the same CDL binding site and their occurrence in *E. coli* membrane proteins. GlpG contains a CDL binding site with two basic residues separated by 75 positions. Site 1 in ROCKET (four positions) is indicated for reference. (**B**) Topology diagram of GlpG indicating the locations of basic (R; arginine, L; lysine) and aromatic (W; tryptophan, Y; tyrosine) residues that interact with CDL. (**C**) GFP-thermal shift assay of GlpG in detergent or in the presence of PE, PC, SM, PG, or CDL. The average fluorescence intensity (FI) indicates the fraction of soluble protein before (unheated) or after heating to 63 °C and removing precipitated protein. Data are normalized against the non-heated control in detergent. All measurements were performed in triplicates (n=3). (**D**) CG-MD-derived Lipid density plots for CDL (left) and POPG (right) around the transmembrane region of GlpG (PDB ID 2IC8) viewed from the cytoplasmic side. Units are number density. CDL, but not POPG, exhibits preferential binding to the R92-K167 site. The backbones of R92, K67, and Y160 are shown as spheres (basic in blue, aromatic in orange). (**E**) nMS spectra of GlpG released from detergent micelles show a 1.4 kDa adduct which can be removed through collisional activation of the protein. (**F**) GlpG proteolytic cleavage rates (specific activity, by normalizing to the GFP signal) for soluble (extramembrane) substrate (KSp63) or transmembrane substrate (KSp96, transmembrane helix shaded) in the presence of PG or CDL. Both lipids increase the cleavage rate for the soluble substrate by approximately 40%. For the transmembrane substrate, addition of PG causes a moderate reduction in cleavage activity, whereas CDL causes near-complete inhibition of cleavage (CDL KSp96 p<0.0001, CDL KSp63 p=0.028, PG KSp96 p=0.0171, and PG KSp63 p=0.0089, all analyzed with two-tailed t-test, n=3).

The online version of this article includes the following figure supplement(s) for figure 4:

**Figure supplement 1.** Analysis of CDL binding sites in *E. coli* proteins and GlpG.

**Figure supplement 2.** Analysis of the stabilizing CDL binding site in the Yeast ADP/ATP carrier Aac2.

presence of different lipids showed that phosphatidyl ethanolamine (PE), phosphatidyl choline (PC), as well as sphingomyelin (SM; which is not found in *E. coli*) had no stabilizing effect, whereas phosphatidyl glycerol (PG) and CDL resulted in near-complete protection form heat-induced precipitation (*Figure 4c*).

When performing CG-MD of GlpG in the model *E. coli* bilayer, we observed that CDL bound exclusively to the predicted site between helix 1 and 4, with an occupancy of 96.2% with an average residence time of 170 ns, but no preferential POPG binding was observed anywhere on the protein (*Figure 4d*, *Figure 4—figure supplement 1*). On the cytoplasmic side, the non-conserved residues W136 and R137 can potentially interact with CDL; however, no increased CDL density was observed (*Figure 4—figure supplement 1*). The data suggest that the stabilizing effect of POPG in the thermal shift assay is due to binding to this site in the absence of CDL. The near-100% occupancy by CDL in the mixed bilayer MD simulations suggests that CDL outcompetes POPG when both lipids are present. To test whether GlpG binds CDL in the membrane, we used nMS analysis. We have previously noted a 1.4 kDa adduct on purified GlpG (*Yen et al., 2022*) and monomeric GlpG with sufficient activation to remove DDM micelle. Collisional activation revealed preferential loss of the adduct as a single species, confirming that it is a single lipid molecule, that is CDL (*Figure 4e*). These findings are in good agreement with the presence of four lipid tails in total, which likely stem from a single CDL in this location in the crystal structure of GlpG (*Figure 4—figure supplement 1*), although these were tentatively modeled as a PA and a PE molecule next to each other in the structure (*Vinothkumar, 2011*). The apparent high specificity for CDL prompted us to investigate the effect of CDL and PG on GlpG protease activity. For this purpose, we used a cleavage assay with fluorescently labeled substrates that represent either a soluble 'extramembrane' substrate (KSp63) or transmembrane helix (KSp96) peptides containing the same GlpG cleavage site (*Poláchová et al., 2023*; *Tichá et al., 2017*). We found that PG and CDL both have a positive effect on the cleavage of the soluble substrate KSp63 compared to detergent-only, with both lipids increasing the cleavage rates by approximately 40% (*Figure 4f*). PG mildly reduced the cleavage rate of the transmembrane substrate KSp96 to approximately 90% compared to detergent-only. Strikingly, CDL caused drastic inhibition of transmembrane substrate cleavage, with only 20% activity remaining (*Figure 4f*). To find out how the lipid binding site, which is located on the cytoplasmic side, can affect substrate access to the active site, which faces the periplasm, we performed CG-MD simulations of the open and closed conformations of GlpG (*Wang and Ha, 2007*). We found that both conformations bound CDL in the same place, which agrees with the observation that CDL affects the flexibility of the linker connecting the N-terminal domain (*Wang and Ha, 2007*). Interestingly, the acyl chains of the tightly bound lipid extend into the gap between helices 2 and 5 (*Figure 4—figure supplement 1*), the lateral gate through which transmembrane substrates access the active site of GlpG (*Wang and Ha, 2007*). Loop 5, whose movement is critical for catalytic activity, remains unaffected, providing a rationale for the selective inhibition of transmembrane substrate cleavage by CDL. Although the biological role of CDL-regulated substrate preferences of GlpG remains to be clarified, our results demonstrate the identification of a functionally relevant CDL binding site in an *E. coli* membrane protein based on the insights from artificial protein design.

## Discussion

In this study, we have systematically investigated the requirements for CDL binding and stabilization of membrane proteins, using the scaffold protein ROCKET as a model system. CDL binding sites have few sequence requirements, which is evident from the fact that ROCKET contains multiple interaction sites solely from statistical distribution of aromatic and charged amino acids at the membrane interface (*Figure 1*). These sites exhibit characteristic features, including electrostatic interactions with the negatively charged lipid headgroups and flanking aromatic residues that align the acyl chains with the hydrophobic transmembrane domain. By introducing mutations at these sites, we can probe their contribution to CDL-mediated stabilization with nMS. To our surprise, we find that although CDL binding sites share well-defined structural features, mutations of these features does not always yield predictable structural effects. Instead, their ability to contribute to lipid-mediated stabilization depends on multiple additional factors. For example, mutations that reduce CDL binding to a high-affinity site can cause redistribution to lower-affinity sites that have, in turn, a more pronounced effect on stability. This appears to be the case for ROCKET*^AAXWA*. Furthermore, mutations that reduce lipid

binding at a low-affinity site may change the local stability of the protein, and results in an increase in lipid-mediated stabilization, as seen in *ROCKET*[R66A]. Lipid interactions are dynamic and heterogeneous, and mutating a well-defined CDL site can give rise to multiple compensatory interactions which in turn affect the local protein environment.

Despite these challenges, the different CDL binding sites in ROCKET demonstrate that commonly observed structural features (the number of headgroup-interaction residues, local flexibility, and the involvement of multiple transmembrane helices) impact lipid-mediated stabilization to different extent. Building upon the insights garnered from ROCKET, we extended our investigation to *E. coli* proteins and identified GlpG as a case study for CDL's impact on function. Previous studies have shown that GlpG conformation is impacted by the surrounding lipids and membrane geometry (*Yen et al., 2022*; *Vinothkumar, 2011*), and CDL has been suggested to affect proteolytic activity (*Urban and Wolfe, 2005*). Our observation that CDL acts as an allosteric activator for the cleavage of soluble substrates while exerting an inhibitory effect on the processing of transmembrane substrates establishes CDL as a regulator of GlpG activity. Similarly, a CDL binding site (W70/R71/R151) in the yeast ATP/ADP carrier Aac2 has the same hallmarks as the GlpG site (*Figure 4—figure supplement 2*). CDL binding to this site connects the flexible N-terminal transmembrane helix to the protein core, resulting in stabilization of the tertiary structure and increased transport activity. nMS shows that the protein retains co-purified CDL (*Senoo et al., 2024*). These findings suggest that the features that mediate lipid stabilization in ROCKET are hallmarks of functionally important lipid binding sites in native membrane proteins.

Although we can identify relatively intuitive features of CDL interaction sites, we find that the connection between lipid binding and stabilization is not clear-cut. For example, one destabilizing mutation increases lipid-mediated stabilization whereas another does not (compare ROCKET[R66A] and ROCKET[A61P]). Furthermore, being able to bind more lipids do not translate to forming more stable complexes (compare ROCKET and ROCKET[AAXWA]). The reasons are likely twofold: Firstly, our approach has methodological limitations, as gas-phase stability is not easily correlated with condensed phase stability. In case of CDL, increasing the number of molecular contacts likely translates to stabilizing effects in both phases. However, charge interactions are relatively strengthened in the gas phase, whereas some hydrophobic contacts will be lost. For example, CDL sites contain aromatic residues (W or Y) close to the ester bonds of the lipids, which likely serve to orient the lipids, but the roles of which have not been examined in ROCKET. Unlike charge interactions with lipid head groups, such subtle contributions are likely distorted by the transfer to the gas phase, making it difficult to confidently assign changes in stability or lipid occupancy. Furthermore, the bulk of the nMS analysis is done using detergents, which have delipidating effects, meaning we will observe fewer lipid binding events and lower occupancy in nMS than in the CG-MD simulations (*Bolla et al., 2020*). Collisional activation required to remove the detergent micelle or lipid vesicle additionally strips away lipids that bind predominantly via hydrophobic interactions.

To avoid this limitation of our MS approach, we have focused here CDL bound via direct headgroup contacts, which can be readily predicted with CG-MD and analyzed with nMS. Secondly, our approach highlights that protein-lipid interactions are complex. The ROCKET scaffold is, *per* design, extremely stable and has no function. Thus, it does not capture the dynamic nature of most membrane proteins, which, in turn, is often related to lipid-mediated regulation (*Landreh et al., 2017*). The key finding of our study is that emulating the architecture of a binding site is only the first step to uncovering the principles of lipid regulation, and that the effect of lipids on the local dynamics are critical to understanding how they shape membrane protein function. Integrated design approaches based not only on static structures but on dynamic models are therefore the key to designing membrane proteins with membrane-specific functions.

In summary, we present a protein design-driven approach to decipher mechanisms by which CDL regulates membrane protein stability. Our findings illuminate critical protein features that are challenging to predict or design de novo. Further integration of MD simulations and nMS with protein engineering could help to overcome some of the challenges identified here, such as the impact of local protein flexibility, and offer a promising pathway to uncover novel CDL binding sites. The findings not only contribute to our understanding of lipid-protein interactions but highlight potential avenues for the design of membrane proteins with tailored stability and function, potentially informing therapeutic strategies targeting membrane protein dysfunctions.

# Materials and methods

Coarse-grained molecular dynamics simulations of ROCKET variants and GlpG. CG systems were built using PDB 6B85 (TMHC4_R) or 2IC8/2NRF (GlpG). For the 6B85 system, two subunits of the heterotetramer were unchanged from the input PDB, whereas the other two were mutated to create the ROCKET$^{AAXWA}$ variant These mutations were added in PyMOL (*Schrödinger, 2015*). Protein atoms were converted to the CG Martini 3 force field (*Souza et al., 2021*) using the martinize method (*Kroon et al., 2023*). Additional bonds of 500 kJ mol$^{-1}$ nm$^{-2}$ were applied between all protein backbone beads within 1 nm. Proteins were built into membranes composed of 10% CDL, 23% POPG, and 67% POPE for ROCKET, or 10% CDL, 10% POPG, and 80% POPE or just 10% POPG and 90% POPE for GlpG. The default CDL Martini 3 parameters were used, corresponding to di-PO tail types. Membranes were built using the insane protocol (*Wassenaar et al., 2015*). All systems were solvated with Martini waters and Na$^+$ and Cl$^-$ ions to a neutral charge and 150 mM. Systems were minimized using the steepest descents method, followed by 1 ns equilibration with 5 fs time steps, then by 100 ns equilibration with 20 fs time steps, before 5 × 10 µs production simulations using 20 fs time steps, all in the NPT ensemble at 323 K with the V-rescale thermostat ($\tau$ =1.0 ps) and semi-isotropic Parrinello-Rahman pressure coupling at 1 bar ($\tau$ =12.0 ps). The reaction-field method was used to model long-range electrostatic interactions. Bond lengths were constrained to the equilibrium values using the LINCS algorithm. Density analyses was performed using the VolMap tool of VMD, with the default settings (*Humphrey et al., 1996*). Lipid binding sites and lipid-residue interactions were determined using the PyLipID package, which provides both occupancy and residence time data (*Song et al., 2022*). Reported occupancy and residence time values are taken analysis of the total simulation time of 5x10 µs. Simulations were run in Gromacs 2022 (*Abraham et al., 2015*; *Abraham et al., 2023*).

## Atomistic molecular dynamics simulations of ROCKET

A post-CG simulation snapshot of ROCKET with bound CDL lipids was converted to an atomistic description using the CG2AT approach (*Vickery and Stansfeld, 2021*) Atoms were described using the CHARMM36m force field (*Huang and MacKerell, 2013*) with TIP3P water. Converted systems were energy minimized using the steepest descents method, and subsequently equilibrated with positional restraints on heavy atoms for 100 ps in the NPT ensemble at 303 K with the V-rescale thermostat ($\tau$ P=1.0 ps) and semi-isotropic Parrinello-Rahman pressure coupling at 1 bar ($\tau$ P=5.0 ps), with a compressibility of 4.5×10−5 bar−1. The Particle-Mesh-Ewald (PME) method was used to model long-range electrostatic interactions. van der Waals (VDW) interactions were cut off at 1.2 nm. Bond lengths were constrained to the equilibrium values using the LINCS algorithm. A production simulation was run to assess the dynamics of the bound lipid, using a 2 fs time step for 530 ns. Simulations were run in Gromacs 2022 (*Abraham et al., 2015*; *Abraham et al., 2023*). A video was made using VMD.

## Gas-phase molecular dynamics simulations of ROCKET variants

In order to assess how lipid binding affects the stability of ROCKET and the ROCKET$^{AAWXA}$ variant in the gas phase, molecular dynamics simulations were performed, in which chains were pulled apart and the force was measured with and without CDL bridging them. The proteins were simulated as homotetramers, using the all-atom models. The chains were truncated to include only the membrane-spanning domain with residues S2-V79, with a neutral C-terminus (COOH) added to residue 79. In aiming to reflect the 16+ charge state, which was observed in nMS experiments, residues D38 and E78 were protonated, while all other titratable residues were given their pKa-based protonation state at pH 7. The addition of eight protons to the truncated model is intended to replicate the distribution of exposed acidic sites on the entire protein, which are likely to become protonated in the experiments. Each variant was simulated without any lipids, as well as with one CDL molecule bound between two helices of adjacent subunits.

Simulations were performed with the GROMACS MD package (*Abraham et al., 2015*), version 2023.3, and the July 2022 version of the CHARMM36 force field (*Best et al., 2012*). The proteins were placed in cubic boxes with sides of 999.9 nm, and the cutoff radii for Coulomb and van der Waals interaction were set to 333.3 nm. This set-up allows the use of the Verlet buffer scheme (*Páll and Hess, 2013*) for neighbour searching and GPU acceleration while avoiding artefacts from the periodicity (*Konermann, 2017*). Virtual sites (*Feroz et al., 2018*) were used for hydrogens, with those for CDL generated using MkVsites (*Larsson et al., 2020*). The four models were equilibrated with a

steepest descent energy minimization until convergence, followed by temperature coupling over 10 ps, using a 0.5-fs time step, and the Berendsen (*Berendsen et al., 1984*) thermostat set to 300 K. For the production simulations, the thermostat was changed to velocity rescale (*Bussi et al., 2007*), retaining the temperature of 300 K. All bonds were constrained with LINCS (*Hess et al., 1997*), using an order of 4 and one iteration, which allowed for a 5-fs time step to be used.

Helices were pulled apart with the center-of-mass pull code. The reference groups included, respectively, the Cα atoms of residues 6–12 and 63–69 on the two chains with which CDL was interacting. The pull coordinate was defined as the distance between the mass centra of the two groups, measured in three-dimensional space. An umbrella potential with a harmonic force constant of 1000 $kJ/mol/nm^2$ and a pull rate of 0.1125 nm/ns was used. Twenty replicate pulling simulations of 50 ns each were performed for each of four models using a different random seed for the initial velocities in each replica. The helix content was computed with the DSSP (*Kabsch and Sander, 1983*) module of MDAnalysis (*Michaud-Agrawal et al., 2011*; *Gowers et al., 2016*).

## Protein engineering of ROCKET mutants

All protein mutagenesis was performed with the Q5 site-directed mutagenesis kit from NEB (E0554). Primers were designed with the NEBaseChanger tool (https://nebasechanger.neb.com/) and synthesized (Eurofins Genomics) with the sequences listed (*Supplementary file 1, table S2*). The original plasmid used as template for the PCR reactions was based on wildtype ROCKET plasmid (*Supplementary file 1, table S3*) and the reactions utilized the Q5 Hot Start High-Fidelity DNA Polymerases with annealing temperature optimized for each individual reaction (*Supplementary file 1, table S3*). PCR products were prepared using kinase, ligase, and DpnI (KDL) mixture, according to the manufacturer protocol, transformed into NEB 5-alfa cells, plated on Luria-Bertani (LB) agar plates containing kanamycin (50 μg/mL) and incubated overnight at 37 °C. Individual colonies were selected for growth in 5 mL LB cultures at 37 °C for 12–16 hr. 1–5 mL of cell pellet was harvested for plasmid extraction with the Monarch Plasmid DNA Miniprep Kit (New England Biolabs; T1010). Mutagenesis was confirmed by sequencing.

## Cloning and expression of ROCKET variants

Synthetic genes with N-terminal histidine tag (either 6-His or 10-His for ROCKET[AAXWA]) were synthesized by Genscript Inc or derived from mutagenesis in a pet26b (+) expression vector (*Supplementary file 1, table S3*). These plasmids were transformed into *E. coli* BL21 (DE3) (New England BioLabs). Selection was carried out on LB agar plates containing kanamycin (50 μg/mL) and incubated overnight at 37 °C. Pre-cultures were grown under the same condition overnight and used to inoculate 400 mL LB media at a 1:100 dilution for protein expression. The expression cultures were incubated at 37 °C until $OD_{600}$ reached 0.8–1.0, at which protein expression was induced with 0.2 mM isopropylthio-β-galactoside (IPTG). Post-induction, the cultures were incubated at 18 °C overnight for protein expression. Cells were harvested by centrifugation at 8000 x *g* for 20 min.

## Purification of ROCKET variants

Cell pellets were homogenized in resuspension buffer (25 mM TRIS pH 8.0, 150 mM NaCl) and subjected to probe sonication. n-Dodecyl-beta-maltoside (DDM) was added to the lysate to a final concentration of 1% (w/v) and the mixture was incubated overnight at 4 °C with shaking. Following solubilization, the solution was centrifuged at 10,000 x *g* for 10 min the supernatant was then filtered through a 0.2 μm syringe filter. The cleared supernatant was applied to a $Ni^{2+}$ Sepharose High Performance column HisTrap (Cytiva) pre-equilibrated with wash buffer (25 mM TRIS pH 8.0, 150 mM NaCl, 30 mM imidazole, and 0.1% DDM). The column was subsequently washed with the same buffer to remove unbound material.

Protein was eluted with an elution buffer (25 mM TRIS pH 8.0, 150 mM NaCl, 300 mM imidazole and 0.1% (w/v) DDM). Elution fractions were collected and analyzed for protein content by SDS-PAGE. Fractions containing the protein of interest were pooled and concentrated using Amicon Ultra-15 centrifugal filter units with a 100 kDa molecular weight cutoff (Merck Millipore) Concurrently, the buffer was changed to remove imidazole, resulting in a final storage buffer (25 mM TRIS pH 8.0, 150 mM NaCl and 0.1% (w/v) DDM) for downstream applications.

## Cryo-EM sample preparation and data acquisition

5 mg/mL ROCKET and ROCKET$^{AAXWA}$ with 100 µM CDL were frozen on Quantifoil 1.2/1.3 Au 300 mesh grids (Quantifoil Micro Tools). 3 µL of sample was applied to each grid which was then blotted for 3 s and plunge-frozen into liquid ethane using FEI Virtobot Mark IV (Thermo Fisher Scientific). Micrographs were collected on Krios G3i electron microscope (Thermo Fisher Scientific) operated at 300 kV equipped with Gatan BioQuantum K3 image filter and a Ceta-D detector. Movies were collected at a nominal ×165,000 magnification, resulting in a pixel size of 0.5076 Å. A total dose of 60 e$^-$/Å$^2$ was used to collect 29 frames over 1 s. The target defocus range was set between –0.6 and –1.8 µm, in steps of 0.2 µm.

## Image processing and model building

Data processing was performed using the RELION 4.0.1 pipeline (*Kimanius et al., 2021*). Motion correction was performed using Relion's own implementation (*Zivanov et al., 2019*) and CTF estimation was done with CtfFind4.1 (*Rohou and Grigorieff, 2015*). For the WT dataset, 2D references from 2D classification of manually picked particles were used for initial autopicking, and the picked particles were used to generate a low-resolution 3D reconstruction that was used and a 3D reference for the final round of autopicking. Due to high similarity in 2D classes from manually picked particles for the WT and MUT5 dataset, the same low resolution 3D reconstruction from the WT processing pipeline was used directly as a 3D reference for autopicking in the MUT5 data set. For both datasets, the auto picked particles were used for an ab initio reconstruction. The particles were then further refined, and the data cleaned using several rounds of 3D classification and 3D auto-refinement, followed by CTF parameters refinement and particle polishing before a final 3D auto-refinement and post-processing. No smaller particles were identified during manual and automated processing.

## Preparation of proteoliposomes for native mass spectrometry

*E. coli* polar lipid extract, with a composition of 67.0% PE, 23.2% PG, 9.8% CDL (Avanti Polar Lipids) was dissolved in a 1:1 mixture of chloroform and methanol. The solvent was then removed under vacuum using a SpeedVac concentrator (Savant SPD1010) until completely dry. The dried lipid film was resuspended in buffer (25 mM TRIS pH 8.0, 150 mM NaCl) and vortexed vigorously to ensure homogeneity. Large unilamellar vesicles (LUVs) were prepared by extrusion though a pair of 0.4 µm polycarbonate membrane. The target protein-to-lipid ration was 1:100 (protein:lipid by weight) and the mixture was incubated at 37 °C for 30 min to allow for protein incorporation into the lipid vesicles. Post-incubation, the samples were dialyzed against 500 mM ammonium acetate, pH 8.0, overnight to facilitate buffer exchange and removal of remaining detergent. The particle size distribution of LUVs and proteoliposomes were monitored by dynamic light scatter (DLS) using a Viscotek model 802 DLS instrument with an internal laser (825–832 nm). Data processing was performed with OmniSIZE2.

## Native mass spectrometry of proteoliposomes

Proteoliposome samples were subjected to ESI nMS were performed on the Q Exactive Ultra-high range (UHMR) mass spectrometer (Thermo Fisher Scientific). The MS capillaries were custom-pulled and coated in-house (*Hernández and Robinson, 2007*). We set the capillary voltage to 1.5 kV and maintained the source temperature at 270 °C. In-source trapping voltage was applied at 300 V to enhance ion desolvation, and the higher-energy collision dissociation (HCD) voltage was set to 200 V to facilitate ion transmission. The ultra-high vacuum pressure within the MS was measured at 6.01x10$^{-10}$ mbar. Data was analyzed using Xcalibur 2.2 (Thermo Fisher).

## Protein preparation for native mass spectrometry

Immediately prior to MS analysis, the purified protein was subjected to size exclusion chromatography(SEC) using a Superdex 200 Increase 10/300 GL column (Cytiva). The detergent exchange process was conducted with native compatible buffer (200 mM ammonium acetate pH 8.0, 0.5% C8E4). The fractions containing the tetrameric state of ROCKET protein was collected for direct analysis with nMS.

## Native mass spectrometry

ESI-MS spectra were recorded on a Waters Synapt G1 wave ion mobility mass spectrometer, modified for high-mass analysis (MS Vision), and equipped with an offline nanospray source. The ESI-MS parameters were set as follows: capillary voltage at 1.5 kV, cone voltage at 100 V, source pressure maintained at 8 mbar, and source temperature regulated at 30 °C. To optimize the detection of protein-detergent complexes, the trap voltage was varied from 90 to 240 V. For assessments of lipid binding ROCKET and ROCKET[AAXWA] variants were prepared in samples with and without 50 µM 16:0 cardiolipin (Avanti Polar Lipid). The nMS setting were the same as above, with a trap voltage of 170 V. For each condition, four protein-lipid mixtures were prepared and measured separately. The mass spectra were analyzed using MassLynx software version 4.1 (Waters), and the intensities of apo- and lipid-bound tetramers were quantified with mMass V3.9.0.

To measure lipid-mediated stabilization, 15 µM ROCKET or ROCKET[MUT] were mixed with 15 µM ROCKET[AAXWA] at an equimolar ratio to achieve comparable intensities for tetrameric species at a collision voltage of 170 V. The mixtures were the supplemented with 16:0 cardiolipin (Avanti Polar Lipid) to a final lipid concentration of 25 µM. nMS setting were consistent with previous experiments and the collision voltage set to 200–220 V to allow optimal detection of unfolded monomers. Four spectra were recorded for each condition (n=4). The mass spectra were analyzed using MassLynx software version 4.1 (Waters), and the intensities of the charge state of the monomers were quantified with mMass V3.9.0.

## Screening of lipid impact on of GlpG-GFP fusion protein stability

GlpG-GFP fusion protein was expressed and purified as described previously (*Nji et al., 2018*). The purified fusion protein was diluted in buffer containing 20 mM Tris-HCl pH 8.0, 150 mM NaCl, 1% (w/v) β-OG and 1% (w/v) DDM to a final concentration of fusion protein of 1 µM, and individual lipids were added to this preparation to a final concentration of 0.3 mg/mL. The used lipids were from Avanti Polar Lipids: 18:1 PE (cat no. 850725 P), 18:1 PG (cat no. 840475 P), 18:1 PC (cat no. 850375 P), 18:1 CDL (cat no. 710335 P) and brain SM (cat no. 860062 P), and DDM was used as a negative control. Samples were incubated at 63 °C for 10 min (the negative control at 4 °C) followed by centrifugation at 20,000 x *g* at 4 °C for 45 min. Fluorescence of the supernatant was measured with TECAN Infinite M1000 spectrophotometer with excitation at 488 nm and emission at 512 nm.

## Determination of protease activity of GlpG

To determine GlpG activity, the GlpG-GFP supernatant collected after centrifugation was diluted 1:10 into 50 mM phosphate buffer pH 7.4, 150 mM NaCl, 0.05% (w/v) PEG 8000, 20% (v/v) glycerol, and 0.05% (w/v) DDM. Lyophilized substrate were dissolved in the same buffer, with further addition of 5% (v/v) DMSO in case of the soluble substrate KSp63 (*Poláchová et al., 2023*), and preincubated at 37 °C. The concentration of substrates in these master mixes were 400 µM for the 'soluble' (extramembrane) substrate KSp63 and 50 µM for the transmembrane substrate KSp96 (*Tichá et al., 2017*). Cleavage reactions were initiated by mixing the enzyme solution and substrate master mix in 1:1 ratio, yielding final concentrations of 200 µM KSp63 and 25 µM KSp96. Protease activity was measured at 37 °C by reading fluorescence intensity continuously every 30 seconds in TECAN Infinite M1000 for 40 min or until the upper detection limit was reached. Excitation was set to 355 nm and emission to 450 nm for the soluble substrate KSp63, or to 335 nm and 493 nm, respectively, for the transmembrane substrate KSp96.

## Acknowledgements

The computations were enabled by resources provided by the National Academic Infrastructure for Supercomputing in Sweden (NAISS) at the PDC Center for High Performance Computing, KTH Royal Institute of Technology, partially funded by the Swedish Research Council through grant agreement (2022–06725). Additional computational resource came from ARCHER and JADE UK National Supercomputing Services, provided by HECBioSim, the UK High End Computing Consortium for Biomolecular Simulation (https://www.hecbiosim.ac.uk), which is supported by the EPSRC (EP/L000253/1). EM data was collected at the Karolinska Institutet 3D-EM facility (https://ki.se/cmb/3d-em). MLA is supported by a VR Research Environment grant (2019–02433). ML is supported by a KI faculty-funded

Career Position, a Cancerfonden Project grant (22–2023 Pj), a VR Starting Grant (2019–01961), and a Consolidator Grant from the Swedish Society for Medical Research (SSMF). KS and JS have been supported by the Operational Programme European Regional Development Fund (no. CZ.02.1.01/0.0 /0.0/16_019/0000729). KS also acknowledges support from the InterCOST programme of the Ministry of Education, Youth and Sports of the Czech Republic (project no. LUC23180). CVR and AOO are supported by the Medical Research Council (MRC) programme grant (MR/V028839/1). DD acknowledges support from Göran Gustafssons Foundation. OA was supported by a doctoral fellowship from Sven och Lilly Lawskis stiftelse. RJH and EL by grants from the Swedish Research Council (2019–02433, 2021–05806) and Swedish e-Science Research Center. LJP and EGM are supported by a project grant from the Swedish Research Council (2020- 04825). PJS acknowledges Wellcome (208361/Z/17/Z), MRC (MR/S009213/1), BBSRC (BB/P01948X/1, BB/R002517/1, BB/S003339/1 and BB/Y003187/1), and the Howard Dalton Centre for funding. PJS. acknowledges Sulis at HPC Midlands+, which was funded by the EPSRC on grant EP/T022108/1, and the University of Warwick Scientific Computing Research Technology Platform for computational access.

## Additional information

### Funding

| Funder | Grant reference number | Author |
|---|---|---|
| Vetenskapsrådet | 2019-02433 | Rebecca J Howard<br>Erik Lindahl<br>David Drew<br>Michael Landreh |
| Vetenskapsrådet | 2019-01961 | Michael Landreh |
| Svenska Sällskapet för Medicinsk Forskning | Consolidator Grant | Michael Landreh |
| Medical Research Council | MR/V028839/1 | Carol V Robinson |
| Vetenskapsrådet | 2021-05806 | Rebecca J Howard<br>Erik Lindahl |
| Wellcome Trust | 10.35802/208361 | Phillip J Stansfeld |
| Medical Research Council | MR/S009213/1 | Phillip J Stansfeld |
| Biotechnology and Biological Sciences Research Council | BB/P01948X/1 | Phillip J Stansfeld |
| Biotechnology and Biological Sciences Research Council | BB/S003339/1 | Phillip J Stansfeld |
| Biotechnology and Biological Sciences Research Council | BB/Y003187/1 | Phillip J Stansfeld |
| Howard Dalton Centre | | Phillip J Stansfeld |
| European Regional Development Fund | Operational Programme CZ.02.1.01/0.0/0.0/16_019 /0000729 | Kvido Strisovsky<br>Jan L Skerle |
| Ministerstvo Školství, Mládeže a Tělovýchovy | InterCOST programme (project no. LUC23180) | Kvido Strisovsky |
| Göran Gustafsson Foundation | | David Drew |
| Sven och Lilly Lawskis Fond för Naturvetenskaplig Forskning | | Olivia Anden |

| Funder | Grant reference number | Author |
|---|---|---|
| Swedish Research Council | 2020- 04825 | Louise J Persson<br>Erik G Marklund |
| Cancerfonden | 22–2023 Pj | Michael Landreh |

The funders had no role in study design, data collection and interpretation, or the decision to submit the work for publication. For the purpose of Open Access, the authors have applied a CC BY public copyright license to any Author Accepted Manuscript version arising from this submission.

## Author contributions

Mia L Abramsson, Conceptualization, Data curation, Formal analysis, Investigation, Methodology, Writing – original draft, Writing – review and editing; Robin A Corey, Conceptualization, Resources, Data curation, Software, Formal analysis, Supervision, Investigation, Methodology, Writing – original draft, Writing – review and editing; Jan L Skerle, Formal analysis, Investigation, Methodology, Writing – review and editing; Louise J Persson, Resources, Data curation, Software, Formal analysis, Investigation, Methodology, Writing – review and editing; Olivia Anden, Resources, Software, Formal analysis, Supervision, Investigation, Writing – review and editing; Abraham O Oluwole, Investigation, Methodology; Rebecca J Howard, Conceptualization, Resources, Supervision; Erik Lindahl, Carol V Robinson, Kvido Strisovsky, Conceptualization, Resources, Supervision, Writing – review and editing; Erik G Marklund, Conceptualization, Resources, Software, Formal analysis, Supervision, Writing – review and editing; David Drew, Conceptualization, Resources, Supervision, Writing – original draft, Writing – review and editing; Phillip J Stansfeld, Conceptualization, Resources, Data curation, Software, Formal analysis, Supervision, Funding acquisition, Investigation, Writing – original draft, Writing – review and editing; Michael Landreh, Conceptualization, Resources, Data curation, Formal analysis, Supervision, Funding acquisition, Investigation, Writing – original draft, Project administration, Writing – review and editing

## Author ORCIDs

Robin A Corey ⓘ https://orcid.org/0000-0003-1820-7993
Louise J Persson ⓘ https://orcid.org/0000-0001-7300-4019
Abraham O Oluwole ⓘ https://orcid.org/0000-0001-8647-4781
Rebecca J Howard ⓘ https://orcid.org/0000-0003-2049-3378
Kvido Strisovsky ⓘ https://orcid.org/0000-0003-3677-0907
Erik G Marklund ⓘ https://orcid.org/0000-0002-9804-5009
David Drew ⓘ https://orcid.org/0000-0001-8866-6349
Michael Landreh ⓘ https://orcid.org/0000-0002-7958-4074

Reviewer #1 (Public review): https://doi.org/10.7554/eLife.104237.3.sa1
Reviewer #3 (Public review): https://doi.org/10.7554/eLife.104237.3.sa2
Author response https://doi.org/10.7554/eLife.104237.3.sa3

# Additional files

## Supplementary files

Supplementary file 1. Supplementary tables 1-3.
MDAR checklist

## Data availability

Cryo-EM density maps of ROCKET and ROCKETAAXWA in detergent micelles have been deposited in the Electron Microscopy Data Bank under accession number EMD-50106 and EMD-50107 respectively. All data are available in the main text or the supplementary materials.

The following datasets were generated:

| Author(s) | Year | Dataset title | Dataset URL | Database and Identifier |
|---|---|---|---|---|
| Abramsson M, Anden O, Howard RJ, Lindahl E, Landreh M | 2024 | Artificial membrane protein TMHC4_R (ROCKET) mutant R9A/K10A/R13A | https://www.emdataresource.org/EMD-50107 | EMDataResource, EMD-50107 |
| Abramsson ML, Anden O, Howard RJ, Lindahl E, Landreh M | 2024 | Artificial membrane protein TMHC4_R (ROCKET) | https://www.emdataresource.org/EMD-50106 | EMDataResource, EMD-50106 |

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
