## [Editor Report · eLife Assessment]

Cardiolipin is known to play an **important** role in modulating the assembly and function of membrane proteins in bacterial and mitochondrial membranes. Here, authors **convincingly** define the molecular determinants of cardiolipin binding on de novo-designed and native membrane proteins combining the coarse-grained molecular dynamics simulation with the state-of-the-art experimental approaches such as native mass spectrometry and cryogenic electron microscopy. The major findings in this study, which are the identification of degenerate cardiolipin binding motifs, the characterization of their dynamic features, and the role in membrane protein stability and activity, will provide much needed insight into the still poorly understood nature of protein-cardiolipin interactions.

---

## [Referee Report · Reviewer #1 (Public review)]

Summary:

The study combines predictions from MD simulations with sophisticated experimental approaches including native mass spectrometry (nMS), cryo-EM, and thermal protein stability assays to investigate the molecular determinants of cardiolipin (CDL) binding and binding-induced protein stability/function of an engineered model protein (ROCKET), as well as of the native *E. coli* intramembrane rhomboid protease, GlpG.

Strengths:

State-of-the-art approaches and sharply focused experimental investigation lend credence to the conclusions drawn. Stable CDL binding is accommodated by a largely degenerate protein fold that combines interactions from distant basic residues with greater intercalation of the lipid within the protein structure. Surprisingly, there appears to be no direct correlation between binding affinity/occupancy and protein stability.

Overall, using both model and native protein systems, this study convincingly underscores the molecular and structural requirements for CDL binding and binding-induced membrane protein stability. This work provides much-needed insight into the poorly understood nature of protein-CDL interactions.

---

## [Referee Report · Reviewer #3 (Public review)]

Summary:

The relationships of proteins and lipids: it's complicated. This paper illustrates how cardiolipins can stabilize membrane protein subunits - and not surprisingly, positively charged residues play an important role here. But more and stronger binding of such structural lipids does not necessarily translate to stabilization of oligomeric states, since many proteins have alternative binding sites for lipids which may be intra- rather than intermolecular. Mutations which abolish primary binding sites can cause redistribution to (weaker) secondary sites which nevertheless stabilize interactions between subunits. This may be at first sight counterintuitive but actually matches expectations from structural data and MD modelling. An analogous cardiolipin binding site between subunits is found in *E. coli* tetrameric GlpG, with cardiolipin (thermally) stabilizing the protein against aggregation.

Strengths:

The use of the artificial scaffold allows testing of hypothesis about the different roles of cardiolipin binding. It reveals effects which are at first sight counterintuitive and are explained by the existence of a weaker, secondary binding site which unlike the primary one allows easy lipid-mediated interaction between two subunits of the protein. Introducing different mutations either changes the balance between primary and secondary binding sites or introduced a kink in a helix - thus affecting subunit interactions which are experimentally verified by native mass spectrometry.

[Editors' note: The reviewers agreed that the authors addressed all reviewer comments adequately and rigorously.]

---

## [Author Response]

The following is the authors’ response to the original reviews

**Public Reviews:**

**Reviewer #1 (Public review):**
Summary:The study combines predictions from MD simulations with sophisticated experimental approaches including native mass spectrometry (nMS), cryo-EM, and thermal protein stability assays to investigate the molecular determinants of cardiolipin (CDL) binding and binding-induced protein stability/function of an engineered model protein (ROCKET), as well as of the native *E. coli* intramembrane rhomboid protease, GlpG.Strengths:State-of-the-art approaches and sharply focused experimental investigation lend credence to the conclusions drawn. Stable CDL binding is accommodated by a largely degenerate protein fold that combines interactions from distant basic residues with greater intercalation of the lipid within the protein structure. Surprisingly, there appears to be no direct correlation between binding affinity/occupancy and protein stability.Weaknesses:(i) While aromatic residues (in particular Trp) appear to be clearly involved in the CDL interaction, there is no investigation of their roles and contributions relative to the positively charged residues (R and K) investigated here. How do aromatics contribute to CDL binding and protein stability, and are they differential in nature (W vs Y vs F)?

Based on the simulations in Corey et al (Sci Adv 2021), aromatic residues, especially tryptophan, appear to help provide a binding platform for the glycerol moiety of CDL which is quite flat. This interaction is likely why we generally see the tryptophan slightly further into the plane of the membrane than the basic residues, where it may help to orient the lipid. Unlike charge interactions with lipid head groups, such subtle contributions are likely distorted by the transfer to the gas phase, making it difficult to confidently assign changes in stability or lipid occupancy to interactions with tryptophan. We have added an explanation of these considerations to the Discussion section (page 13, last paragraph).

(ii) In the case of GlpG, a WR pair (W136-R137) present at the lipid-water on the periplasmic face (adjacent to helices 2/3) may function akin to the W12-R13 of ROCKET in specifically binding CDL. Investigation of this site might prove to be interesting if it indeed does.

Thank you for the suggestion. In our CG simulations, we don’t see significant CDL binding at this site, likely because there is just a single basic residue. We note that there is a periplasmic site nearby with two basic residues (K132+K191+W125) with a higher occupancy, however still far lower than the identified cytoplasmic site. In general, periplasmic sites are less common and/or have lower affinity which may be related to leaflet asymmetry (Corey et al, Sci Adv 2021). We added the CDL density plot for the periplasmic side to Figure S7 and noted this on page 9, next-to-last paragraph.

(iii) Examples of other native proteins that utilize combinatorial aromatic and electrostatic interactions to bind CDL would provide a broader perspective of the general applicability of these findings to the reader (for e.g. the adenine nucleotide translocase (ANT/AAC) of the mitochondria as well as the mechanoenzymatic GTPase Drp1 appear to bind CDL using the common "WRG' motif.)

Several confirmed examples are presented in Corey et al (Sci Adv 2021), the dataset which we used to identify the CDL site in GlpG. So essentially, our broader perspective is that we test the common features observed in native proteins in an artificial system. While it is not clear how a peripheral membrane protein like Drp1 fits into this framework, the CDL binding sites in ANTs indeed have the same hallmarks as the one in GlpG (Hedger et al, Biochemistry 2016). We recently contributed to a study demonstrating that the tertiary structure of ANT Aac2 is stabilized by co-purified CDL molecules, underscoring the general validity of our findings (Senoo et al, EMBO J 2024). We have added this information to the discussion, pg 12, third paragraph, and added a figure (S8, see below) to highlight the architecture of the Aac2-CDL complex.

Overall, using both model and native protein systems, this study convincingly underscores the molecular and structural requirements for CDL binding and binding-induced membrane protein stability. This work provides much-needed insight into the poorly understood nature of protein-CDL interactions.

We thank the reviewer for the positive assessment!

**Reviewer #2 (Public review):**
Summary:The work in this paper discusses the use of CG-MD simulations and nMS to describe cardiolipin binding sites in a synthetically designed, that can be extrapolated to a naturally occurring membrane protein. While the authors acknowledge their work illuminates the challenges in engineering lipid binding they are able to describe some features that highlight residues within GlpG that may be involved in lipid regulation of protease activity, although further study of this site is required to confirm it's role in protein activity.CommentsDiscrepancy between total CDL binding in CG simulations (Fig 1d) and nMS (Fig 2b,c) should be further discussed. Limitations in nMS methodology selecting for tightest bound lipids?

We thank the reviewer for pointing out that this needs to be clarified. We analyze proteins in detergent, which is in itself delipidating, because detergent molecules compete with the lipids for binding to the protein, an effect that can be observed in MS (Bolla et al, Angew Chemie Int. Ed. 2020). Native MS of membrane proteins requires stripping of the surrounding lipid vesicle or detergent micelle in the vacuum region of the mass spectrometer, which is done through gentle thermal activation in the form of high-energy collisions with gas molecules. Detergent molecules and lipids not directly in contact with the protein generally dissociate easier than bound lipids (Laganowsky et al, Nature 2014), however, the even loosely bound lipids can readily dissociate with the detergent, artificially reducing occupancy. The nMS data is therefore likely biased towards lipids bound tightly (e.g. via electrostatic headgroup interactions), however, these are the lipids we are interested in, meaning that the use of MS is suitable here. We have noted this in the Discussion, last paragraph on page 12.

Mutation of helical residues to alanine not only results in loss of lipid binding residues but may also impact overall helix flexibility, is this observed by the authors in CG-MD simulations? Change in helix overall RMSD throughout simulation? The figures shown in Fig.1H show what appear to be quite significant differences in APO protein arrangement between ROCKET and ROCKET AAXWA.

For most of the study, we use CG with fixed backbone bead properties as well as an elastic network to maintain tertiary structure. This means that a mutation to alanine will have essentially no impact on the stability of the helix or protein in general in the CG simulations in the bilayer. It should be noted that Figure 1H shows snapshots from atomistic gas phase simulations with pulling force applied (see schematic in Figure 1F, as well as Figure S1 for ends-point structures), where we naturally expect large structural changes due to unfolding. We have analyzed the helix content in the gas-phase simulations and see that helix 1 in ROCKET unwinds within 10 ns but stays helical ca. 10 ns longer when bound to CDL. The AAWXA mutation stabilizes the helical conformation independently of CDL binding, but CDL tethers the folded helix closer to the core (see Figure 1 G and H). We have added this information to the results section and the plot below to Figure S2.

CG-MD force experiments could be corroborated experimentally with magnetic tweezer unfolding assays as has been performed for the unfolding of artificial protein TMHC2. Alternatively this work could benefit to referencing Wang et al 2019 "On the Interpretation of Force-Induced Unfolding Studies of Membrane Proteins Using Fast Simulations" to support MD vs experimental values.

We apologize for the confusion here. The force experiments are gas-phase all-atom MD. The simulations show that the protein-lipid complex has a more stable tertiary structure in the gas phase. Since these are gas-phase simulations, they cannot be corroborated using in-solution measurements. Similarly, the paper by Wang et al is a great reference for solution simulations, however, to date the only validations for gas-phase unfolding come from native MS.

Did the authors investigate if ROCKET or ROCKETAAXWA copurifies with endogenous lipids? Membrane proteins with stabilising CDL often copurify in detergent and can be detected by MS without the addition of CDL to the detergent solution. Differences in retention of endogenous lipid may also indicate differences in stability between the proteins and is worth investigation.

We have investigated the co-purification of the ROCKET variants and did not observe any co-purified lipids (see Figure S4) which we clarified in the results section (page 5, third paragraph) now. We previously showed that long residence times in CG-MD are linked to the observation of co-purified lipids, because they are not easily outcompeted by the detergent (Bolla et al, Angew Chemie Int. Ed. 2020). In CG-MD of ROCKET, we see that although the CDL sites are nearly constantly occupied, the CDL molecules are in rapid exchange with free CDL from the bulk membrane. For MS, all ROCKET proteins were extracted from the *E. coli* membrane fraction with DDM, which likely outcompetes CDL. This interpretation would explain why we see significant CDL retention when the protein is released from liposomes, but not when the protein is first extracted into detergent. For GlpG, CDL residence times in CG-MD are longer, which agrees with CDL co-purification. Similarly, there is clearly an enrichment of CDL when the protein is extracted into nanodiscs (Sawczyc et al, Nature Commun 2024).

Do the AAXWA and ROCKET have significantly similar intensities from nMS? The AAXWA appears to show slightly lower intensities than the ROCKET.

We did not observe a significant difference, however, in most spectra, the AAXWA peaks have a lower intensity than those of the other variants (see *e.g.* Figure S5). While this could be batch-to-batch variations, there may be a small contribution from the lower number of basic residues (see Abramsson et al, JACS au 2021). However, there is an excess of basic residues in the soluble domain of ROCKET, so this interpretation is speculative.

Can the authors extend their comments on why densities are observed only around site 2 in the cryo-em structures when site 1 is the apparent preferential site for ROCKET.

We base the lipid preference of Site 1 > Site 2 on the CG MD data, where we see a higher occupancy for site 1. At the same time, as noted in the text, CDL at both sites have rather short residence times. When the protein is solubilized in detergent, these times can change, and lipids in less accessible sites (such as cavities and subunit interfaces) may be subject to a slower exchange than those that are fully exposed to the micelle (Bolla et al, Angew Chemie Int. Ed. 2020). We speculate that this effect may favor retaining a lipid at site 2. Furthermore, site 1 is flexible, with CDL attaching in various angles while site 2 has more uniform CDL orientations (see CDL density plot in Figure 1D). EM is likely biased towards the less flexible site. Notably, the density is still poorly defined, so it is possible that a more variable lipid position in site 1 would not yield a notable density at all. We have added this information to the Results section (page 5, second paragraph).

The authors state that nMS is consistent with CDL binding preferentially to Site 1 in ROCKET and preferentially to Site 2 in the ROCKET AAXWA variant, yet it unclear from the text exactly how these experiments demonstrate this.

As outlined in the previous answer, we base our assessment of the sites on the CG MD simulations. There, we note that CDL binds predominantly to site 1 in ROCKET and predominantly to site 2 in AAXWA, however, the overall occupancy is lower in AAXWA than in Rocket, meaning fewer lipids will be bound simultaneously in that variant. The nMS data show CDL retention by both variants when released from liposomes, but the AAXWA has lower-intensity CDL adduct peaks (Figure 2B, C). We interpret this that both have CDL sites, but in the AAXWA variant, the sites have lower occupancy. We agree that this observation does not demonstrate that the CG MD data are correct, however, it is the outcome one expects based on the simulations, so we described it as “consistent with the simulations”. We have rephrased the section to make this clear.

As carried out for ROCKET AAXWA the total CDL binding to A61P and R66A would add to supporting information of characterisation of lipid stabilising mutations.

We considered this possibility too. Unfortunately, the mass differences between A61P / R66A and AAXWA are slightly too high to unambiguously resolve CDL adducts of each variant, as the 1st CDL peak of AAWXA partially overlaps with the apo peak of A61P or R66A.

Did the authors investigate a double mutation to Site 2 (e.g. R66A + M16A)?

While designing mutants, we tested several double mutants involving the basic residues that bind the CDL headgroups (e.g. R66 + AAWXA) but found that they could not be purified, probably because a minimum of positive residues at the N-terminus is required for proper membrane insertion and folding. M16 is an interesting suggestion, but wasn’t considered because the more subtle effects of non-charged amino acids on CDL binding may be lost during desolvation (see also our response to Comment (i) from reviewer 1).

Was the stability of R66A ever compared to the WT or only to AAXWA?

Some of the ROCKET mutants have very similar masses that cannot be resolved well enough on the ToF instrument. While the R66-WT comparison is possible, we would not be able to compare it to R61P or D7A/S8R. To avoid three-point comparisons, we selected AAXWA as the common point of reference for all variants.

How many CDL sites in the database used are structurally verified?

At the time, 1KQF was the only verified *E. coli* protein with a CDL resolved in a high-resolution structure. The complex was predicted accurately, see Figure 6A in Corey et al (Sci Adv 2021), as were several non-*E. coli* complexes.

The work on GlpG could benefit from mutagenesis or discussion of mutagenesis to this site. The Y160F mutation has already been shown to have little impact on stability or activity (Baker and Urban Nat Chem Biol. 2012).

We thank the referee for their excellent suggestion. While Y160F did not have a pronounced effect, the other 3 positions of the predicted CDL binding site in GlpG have not been covered by Baker and Urban. Looking at sequence conservation in GlpG orthologs, manually sampling down to 50% identity (~1300 sequences in Uniprot) shows that Y160 and K167 are conserved, R92 varies between K/R/Q, whereas W98 is not conserved. The other (weak) site cited above (K132 and K191) is not conserved. A detailed investigation of how the conserved residues impact CDL binding and activity is already planned for a follow up study focusing on GlpG biology.

**Reviewer #3 (Public review):**
Summary:The relationships of proteins and lipids: it's complicated. This paper illustrates how cardiolipins can stabilize membrane protein subunits - and not surprisingly, positively charged residues play an important role here. But more and stronger binding of such structural lipids does not necessarily translate to stabilization of oligomeric states, since many proteins have alternative binding sites for lipids which may be intra- rather than intermolecular. Mutations which abolish primary binding sites can cause redistribution to (weaker) secondary sites which nevertheless stabilize interactions between subunits. This may be at first sight counterintuitive but actually matches expectations from structural data and MD modelling. An analogous cardiolipin binding site between subunits is found in *E. coli* tetrameric GlpG, with cardiolipin (thermally) stabilizing the protein against aggregation.

“It’s complicated” We could not have phrased the main conclusions of our study better.

Strengths:The use of the artificial scaffold allows testing of hypothesis about the different roles of cardiolipin binding. It reveals effects which are at first sight counterintuitive and are explained by the existence of a weaker, secondary binding site which unlike the primary one allows easy lipid-mediated interaction between two subunits of the protein. Introducing different mutations either changes the balance between primary and secondary binding sites or introduced a kink in a helix - thus affecting subunit interactions which are experimentally verified by native mass spectrometry.Weaknesses:The artificial scaffold is not necessarily reflecting the conformational dynamics and local flexibility of real, functional membrane proteins. The example of GlpG, while also showing interesting cardiolipin dependency, illustrates the case of a binding site across helices further but does not add much to the main story. It should be evident that structural lipids can be stabilizing in more than one way depending on how they bind, leading to different and possibly opposite functional outcomes.

We share the reviewer’s concern, as we clearly observe that TMHC4_R does not have the same type of flexibility as a natural protein. We find that by introducing flexibility, we start to see CDL-mediated effects. To test the valIdity of our findings from the artificial system, we apply them to GlpG. In response to a suggestion from Reviewer 1, we compared the findings to Aac2, and found that its stabilizing CDL site closely resembles that in GlpG (see new Figure S8).

**Recommendations for the authors:**

**Reviewer #1 (Recommendations for the authors):**
Minor comments:There are a number of typos/uncorrected statements in the text.i) The last sentence of the Abstract appears to be an uncorrected mishmash of two.ii) Line 66: "protects" should be just "protect"iii) Line 75: Sentence appears to be incomplete. "...associated changes in protein stability." The word "stability" is missing.

We have made these changes.

iv) Fig. 2E. Are the magenta and blue colors inverted for variants 1 and 2?

No, the color is correct. greater stabilization of the blue tetramer (AAXAW) compared to WT (purple) will lead to fewer blue monomoers than purple monomers in the mass spectrum.

v) Line 274: the salt bridge should be between R8-E68.

We have corrected this.

vi) Lines 350-354 (final sentence of the paragraph): The sentence does not read well (especially with the double negative element). Please reconstruct the sentence and/or break it into two.

We have split the sentence in two.

Suggestions:(i) While aromatic residues (in particular Trp) appear to be clearly involved in the CDL interaction, there is no investigation of their roles and contributions relative to the positively charged residues (R and K) investigated here. How do aromatics contribute to CDL binding and protein stability, and are they differential in nature (W vs Y vs F)?

See our response to comment (i) from reviewer 1. In short, subtle contribution to lipid interactions (such as pi stacking with Trp or Tyr) will likely be lost during transfer to the gas phase. However, see also our response to the last comment from reviewer 2, we plan to use solution-phase activity assays to investigate the effect of Trp on CDL binding to Glp. However, this is beyond thes cope oif the current study.

(ii) In the case of GlpG, a WR pair (W136-R137) present at the lipid-water on the periplasmic face (adjacent to helices 2/3) may function akin to the W12-R13 of ROCKET in specifically binding CDL. Investigation of this site might prove to be interesting if it indeed does.

We added the CDL density plot for the periplasmic side to Figure S7 and discuss further sites in GlpG in the Discussion section. See response to point (ii) above for details.

**Reviewer #2 (Recommendations for the authors):**
Minor comments- Typo in abstract line 39-40- Typo in figure legend of Fig 1 line 145- Typo in line 149, missing R66 in residues shown as sticks description- Lines 165-167 could benefit from describing what residues are represented as sticks

We have made these changes.

- Line 263 should refer to the figure where the tetrameric state was not affected by this mutation.

The full spectrum of the A61P mutant is not included in the figure, hence there is no reference,

- Addition of statistics to Fig. 4F ?

We have added significance indicators to the graph and information about the statistics to the legend.

**Reviewer #3 (Recommendations for the authors):**
Minor issuesl39: rewrite

We have made these changes.

l60: provide evidence for what is presented as a general statement - cardiolipins might also regulate function without affecting oligomeric state, e.g. MgtA

This is a good point, we have added references to two examples where CDL work without affecting oligomerization (MtgA, Weikum et al BBA 2024, and Aac2, Senoo et al, EMBO J 2024).

l74: not every functional interaction comes with a thermal shift

We use thermal shift as a proxy because it indicates tight interactions, even if they may not be functional. We have made this distinction clearer in the text.

l78: this is true for electrostatic interactions such as are at play here, but not necessarily for hydrophobic onesl133: in what direction is the pulling force applied - the figure seems to suggest diagonally?

The pull coordinate is defined as the distance between the centers of mass of the two helices. The direction of the pull coordinate in Cartesian coordinate space is thus not fixed.

fig 1f, l159: "dissociating" meaning separation of subunits? the placement of the lipid within one subunit would not suggest that intermolecular interactions are properly represented here, please clarify

The lipid placement in the schematic is not representative since the lipid occupies different spaces in WT and AAXWA, we have noted this in the legend. Regarding line 159, “Dissociation” is not strictly correct, since the measure the force to separate helix 1 and 2, *i.e.* unfolding. We have changed the wording to “unfolding”.

l173: was there any evidence in EM data for monomers or smaller oligomers?

No smaller particles were identified by visual inspection or in the particle classes. We have noted this in the methods section.

l203: were tetramer peaks isolated separately for CID?

C8E4 can cause some activation-dependent charge reduction, which could allow some tetramers to “sneak out” of the isolation window. We used global activation without precursor selection which subjects all ions to activation.

fig 2c: can you indicate the 3rd lipid binding as it seems to be in the noise

We can unambiguously assign the retention of three CDL molecules for 17+ charge state only, and clarified this in the legend to Figrue 2.

fig3: can you pls clarify what is meant by stabilization here - less monomer in case A means a more stable oligomer, but "A > B" should lead to ratios < 50%. This does not help with understanding what "stabilization" means in panels c-f, please define what the y axis means for these. Please also explain the bottom panels (side view) in each case, what do the dots represent?

We apologize for the oversight of not explaining the side views, we have added a legend. The schematic in panel A is correct (compare the schematic in Figure 2E). If tetramer A (blue) is stabilized by CDL more than tetramer B “CDL stabilization A>B”, there will be fewer monomers ejected from A. If there is less A in the presence of CDL, then the ratio of B/(B+A) will go up.

It is not very clear what consequences the kink introduced by proline has for intra- vs. intermolecular interactions - the cartoons don't help much here

We agree, the A61P impact on the structure is subtle. The small kink it introduces is not really visible in the top view, and hence, we tried to emphasize this in the side view. We have clarified the meaning of the side view schematics in the legend.

l360: is that an assumption made here or is there evidence for displacement? native MS could potentially prove this.

This is an assumption based on the fact that we see very little binding of POPG in the mixed bilayer CG-MD. We have clarified this in the text. Measuring this with MS is an interesting idea, but we have no direct measurement of displacement, since addition of CDL and POPG to the protein in detergent would result in binding to other sites as well.

fig 4d: there is not much POPG density visible at all - why is that?

Both plots use the same absolute scale. There is simply much less POPG binding compared to CDL.

fig 4e: is this released protein already dissociated into monomers due to denaturation or excessive energy (CID product) - please comment.

The CID energy for the spectrum in Figure 4E was selected to show partial dissociation and monomer release at higher voltages (220V in this case). At lower voltages (150V-170V) we do not observe dissociation in C8E4, see Figure S4A.

l363: pls comment on the apparent discrepancy between single lipid binding and double density

We added a clarifying sentence regarding the double lipids. The density seen in the published structure is of four lipid tails next to each other, which is what one would expect for a CDL. Since the CDL could not be resolved unambiguously, two phospholipids with two acyl chains each were modeled into the density instead. Our MS and MD data strongly suggests that the density stems from a single CDL.